

# Supercooled Drizzle Development in Response to Semi-Coherent Vertical Velocity Fluctuations Within an Orographic Layer Cloud

Adam Majewski[1], Jeffrey R. French[1]

[1]Department of Atmospheric Science, University of Wyoming, Laramie, 82070, USA

*Correspondence to*: Adam Majewski (amajewsk@uwyo.edu)

**Abstract.** Observations of super-cooled liquid water are nearly ubiquitous within wintertime, orographic layer clouds over the intermountain west; however, observations of regions containing super-cooled drizzle drops (SCDDs) are much rarer and the factors controlling SCDD development and location less well understood. As part of the Seeded and Natural Orographic Wintertime clouds—the Idaho Experiment (SNOWIE) goal of improving understanding of natural cloud structure, this study

examines the role of fine-scale (sub-kilometer) vertical velocity fluctuations on the microphysical evolution and location of SCDDs within the observed mixed-phase, wintertime orographic clouds from one research flight of SNOWIE.

This flight saw SCDDs develop in an elevated, postfrontal layer cloud with cold cloud tops (T < -30 °C)—containing low number concentrations of both ice ($N_{ice} < 0.5$ L$^{-1}$) and droplets ($N_{cld} < 30$ cm$^{-3}$). Regions of supercooled drizzle at flight level extended more than a kilometer along the mean wind direction and were first located at and below layers of semi-coherent

vertical velocity fluctuations (SCVVFs) embedded within the cloud. The microphysical development of SCDDs in this environment is catalogued using size and mass distributions derived from in-situ probe measurements. Regions corresponding to hydrometeor growth are determined from radar reflectivity profiles retrieved from an airborne W-band cloud radar. Analysis suggests that SCVVF layers (e.g. from K-H waves) are associated with local SCDD development in response to the kinematic perturbation pattern. This drizzle development and subsequent growth by collision-coalescence is inferred from vertical

reflectivity enhancements (-20 dBZ/km), with drizzle production confirmed by in-situ measurements within one of these vertical velocity fluctuation layers. The SCDD production and growth occurs embedded within cloud over shallow (km or less) layers before transitioning to drizzle production at cloud top further downwind, indicating that wind shear and resultant vertical velocity fluctuations may be more important for SCDD development than cloud top broadening mechanisms in the orographic (or similarly sheared) cloud environment(s).

## 1 Introduction

Over the last forty years, there have been numerous field campaigns either directly or indirectly examining mixed-phase, orographic layer clouds (Hobbs, 1975; Cooper and Saunders, 1980; Heggli and Reynolds, 1985; Rasmussen et al., 1992; Ikeda et al., 2007; Rosenfeld et al., 2013). At cloud-top temperatures between 0 and -20 °C, these clouds frequently contain extensive regions of Supercooled Liquid Water (SLW), especially near cloud top, making such clouds a prime meteorological


environment for aircraft icing (Ashenden et al., 1996; Marwitz et al., 1997). Supercooled Drizzle Drizzle Drops (SCDDs) are the 50-500 μm supercooled drops which have appreciable (0.1-2 m/s) fall velocities relative to cloud droplet (D < 50 μm) motions and consequently grow rapidly in diameter via collision coalescence (dD/dt ~ exp(t); Lamb and Verlinde, 2011). This study aims to catalogue the effect of local, kilometer-scale kinematic perturbation patterns on the development and location of SCDDs for one such mixed-phase cloud system.

Most recent climatologies (Rauber et al., 2000; Bernstein et al., 2007) describe SCDD development as occurring predominantly by collision-coalescence growth in completely supercooled liquid clouds. Studies explicitly examining the microphysical development of SCDDs with in-situ aircraft data confirmed the primacy of the collision-coalescence growth mechanism (Cober et al., 2001) as opposed to the "classical" mechanism—which sees ice hydrometeors melt as they fall through an embedded warm layer (T > 0 °C) before subsequent supercooling as fully melted drizzle drops. Wintertime orographic layer clouds are

frequently too shallow and too cold (outside of cold air damming events on the east coast) to support a warm nose (Rauber et al., 2000)—therefore the climatologies suggest that collision coalescence is the dominant SCDD development mechanism in the clouds of interest in this study.

Collision-coalescence growth is favored in clouds with low cloud droplet number concentrations. For clouds with similar condensate supply rates, populations of fewer droplets will reach condensational "bottleneck" (D ~ 30-40 μm) sizes faster than

more numerous droplets. For this reason, clouds formed in clean air masses (i.e. lower numbers of CCN) or in less vigorous updrafts (where S* is nearer 1 and fewer CCN are activated) are kinetically favored for drizzle formation (Freud and Rosenfeld, 2012). In agreement, the conditions of limited CCN abundance and gradual ascent are linked to high frequency of SCDD formed via collision-coalescence at a climatologic scale (Rauber et al., 2000; Bernstein et al., 2007). Regions which see shallow clouds form from warm, moist air gradually lifted over an arctic cold front or orography frequently see SCDD formation—

faster and more extensively if the clouds form in clean, maritime air masses (Rasmussen et al., 2002). A region that has uplift mechanisms in both orography and surface frontal passage, as well as the required cloud level moisture supply, is the American InterMountain West (IMW) during the winter storm season.

The presence and amount of ice provides another precondition for SCDD development in mixed phase orographic clouds, as the (bulk) ice phase typically acquires mass more rapidly than the liquid phase owing to both an increased diffusional vapor

pressure gradient ($e_{si} < e_i$) and increased individual linear growth rates due to crystal geometry. This places an upper limit on active ice nucleating particle (INP) and ice crystal number for SCDD formation, else ice will more rapidly scavenge the available vapor and cloud water, inhibiting growth of cloud droplets to drizzle sizes (Rasmussen et al., 2002; Geresdi and Rasmussen, 2005). A byproduct is that SCDD observations are infrequent in clouds with cloud tops colder than -15 °C, with few if any observations of SCDD formation found in the literature with cloud tops colder than -23 °C. In the shallow orographic

layer clouds of interest to this study, cloud top temperatures are typically warmer than -20 °C when not part of a deeper precipitating frontal structure, limiting natural primary ice nucleation.

        Collision-coalescence initiation and growth further depends on broadening mechanisms for the largest bottleneck droplets to begin collection of smaller droplets in the population (via fall speed separation). Steady condensational growth





alone is responsible for distribution narrowing around the 40 μm bottleneck diameter, so drop size distribution (DSD)
broadening mechanisms (e.g. turbulent or isobaric mixing, eddy hopping, etc.) are necessary to provide the differential fall
speed conducive to collision-coalescence onset and subsequent rapid collectional growth. Observational results of SCDDs
formed in clouds with greater droplet number ($N_{cld}$ > 100 cm$^{-3}$) indicated that layers of cloud top shear were correlated with
vertical location of drizzle development in cloud, presumably due to this mechanism (Pobanz et al., 1994). Shear-induced
turbulent mixing (especially at cloud top) is thought to be responsible for relatively rapid drop size distribution broadening
(Grabowski and Abade, 2017). Any isobaric mixing of different temperature parcels near the cloud boundary (e.g. with a
strong cloud top inversion) are expected to further accelerate this process.

Supersaturation history provides an analytical framework for understanding several of these broadening mechanisms (e.g.
vertical velocity fluctuations, turbulent eddy hopping, mixing events, etc.) which may be responsible for the rapid spectral
broadening and subsequent collision coalescence behavior in warm stratiform clouds (Cooper et al., 1989; Korolev and Mazin,
1993; Politovitch and Cooper, 1994; Korolev, 1995). For instance, Korolev found that when modeled cloud parcels are
subjected to repeated vertical velocity fluctuations, drop size distributions broaden and may even see a second, small-diameter
droplet mode develop from interstitial CCN activation (hereafter, secondary droplet activation). Turbulence and wave motions
were both suggested as possible meteorological sources for these vertical velocity fluctuations, but the lack of parcel-following
in-situ measurements made validating these behaviors an observational challenge (Pobanz et al., 1994).

Whatever the development mechanism, studies have reported SLW in orographic mixed-phase clouds across the entire IMW
region (Cooper and Saunders, 1980; Rauber and Grant, 1986; Rauber, 1992; Rosenfeld et al., 2013), with amount and spatial
extent decreasing primarily due to number of upstream barriers (Hindman, 1986; Saleeby et al., 2011). Observational and
modeling results confirm that the frequency of SCDD development and collision-coalescence activity within larger SLW
pockets is linked to both low CCN and INP concentrations (Rasmussen et al., 2002; Rosenfeld et al., 2013). The comprehensive
climatology of Bernstein et al. (2007) agreed with the gradient expected based on earlier case studies, with the highest SCDD
frequency extending along the Pacific coastal barrier mountains (in WA, OR, and CA), otherwise showing a decreasing SCDD
frequency with distance inland from the Pacific across the IMW. This reaffirms the expected link between climatologically
clean air masses and SCDD formation (Rauber et al., 2000), however observations (Korolev and Isaac, 2000) and models
(Rasmussen et al., 2002) demonstrate the possibility of SCDD development in *both* maritime and continental air masses given
low enough ice number and active cloud top broadening mechanisms. Rosenfeld et al. (2013) clarified this aerosol-precipitation
relationship for the region, demonstrating that frequent passage of maritime airmasses (i.e. low CCN and INP concentration)
is associated with a higher frequency of SCDD development, greater in-cloud SCDD spatial extent, and persistence of SCDD
to more extreme thermodynamic conditions (e.g. cloud top temperature < -20 °C) relative to continental air masses. Modeling
results have confirmed that for maritime (continental) air masses, freezing drizzle development is faster (slower) due to lesser
(greater) CCN concentrations and occurs over a deeper (shallower) layer near cloud top; however, both situations require ice
crystal concentrations less than about 0.08 L$^{-1}$, at least in models (Rasmussen et al., 2002). According to the expectations of



both the IMW climatology and the drizzle formation mechanism, understood from modeling and observations, freezing drizzle frequency and in-cloud spatial extent is expected to decrease with distance from the Pacific across the IMW.

This study examines an individual case from a field campaign located in southwest Idaho that saw SCDD development in an
winter orographic cloud system despite cold cloud tops (T < -20 °C) which are typically associated with more active ice nucleation and more abundant natural ice (DeMott et al., 2010). The persistently low droplet number concentrations (75th percentile of $N_{CDP}$ cloud observations below 50 cm$^{-3}$ for 12 of 23 flights) and frequent SCDD observations (13 of 23 flights) (Tessendorf et al., 2018) inspired this analysis and seem consistent with the climatological maxima of wintertime SCDD frequency that stretches from the coastal barrier mountains into Idaho (Bernstein et al., 2007). The analysis focuses on the
spatial kinematic patterns and their effect on the liquid phase precipitation development in these mixed phase clouds.

## 2 Study Area and Data

The Seeded and Natural Orographic Wintertime clouds—the Idaho Experiment (SNOWIE) was designed to observe and analyze the evolving wintertime orographic cloud structure in a series of prescribed airborne cloud seeding experiments (Tessendorf et al., 2018). As part of this process, it was necessary to establish the evolution of the natural cloud structure and
microphysics as a baseline for evaluating cloud seeding effects. A separate objective was to use the extensive dataset and state of the art measurements to arrive at new insights towards understanding the natural cloud structure, microphysical evolution, and precipitation patterns, independent of cloud seeding effects. Understanding how fine scale (km or less) dynamical processes impact the cloud microphysical development and spatial distribution, amount, and phase of observed precipitation in these clouds is on the forefront of observational work undertaken in the remote sensing and cloud microphysics communities
(e.g. Houze and Medina, 2005) and provides valuable insight to cloud modeling and microphysical parameterizations.

To characterize and describe the development of precipitation hydrometeors (e.g. SCDDs) at flight level requires knowledge of the instantaneous cloud hydrometeor spectrum, current thermodynamic and dynamic conditions that govern the development of this spectrum, and the spatial variability of these parameters. To catalogue cloud structure and precipitation evloution, the University of Wyoming King Air (UWKA) research aircraft—equipped with remote profiling radar, cloud
probes, temperature and humidity sensors, and a gust probe—repeated fixed flight legs oriented along the mean wind direction through cloud (Fig. 1), at as low an altitude as practical. UWKA legs were anchored above the Packer John (PJ, see Fig. 1) ground site to recurrently sample the same spatial cross sections through the evolving orographic cloud structure, often between the -10 to -15 °C isotherms. Bulk thermodynamic and dynamic atmospheric conditions were characterized by soundings launched at Crouch, ID (KCRH, Fig. 1) before (and during) each flight. Legs were generally no longer than 100 km, with the
western end located over the Payette Valley and the eastern end over the Sawtooth Mountains.

SNOWIE utilized the W-Band Wyoming Cloud Radar to document the orographic cloud structure above and below flight level and provide context for the in-situ cloud microphysics measurements (Vali et al., 1998; Wang and Geerts, 2003; Wang et al., 2012). Previous studies demonstrated that the WCR could resolve fine scale details of orographic clouds (~30 m spatial



resolution), observing aspects of their dynamical and microphysical structure technologically impossible in previous decades

(Aikins et al., 2016). The 95 GHz frequency of the WCR is sensitive to cloud droplets and drizzle in the Rayleigh regime, with Mie effects starting at around 600 µm and reflectivity increasing monotonically with diameter up to millimetric sizes (D > 0.95 mm). Radar reflectivity for volumes containing even large drizzle drops was therefore dominated by the contribution of the largest drops, as no SCDD drizzle drops were observed with diameters greater than about 0.5 mm over the course of SNOWIE based on particle images captured by in situ probes. Additional pulse-pair Doppler velocity measurements captured

the near-vertical, reflectivity-weighted motions of the distributed hydrometeor targets.

In situ probes on the UWKA measured cloud hydrometeors from a few microns to several millimeters in size (Table 1). Two probe types were used to collect these data—a forward scattering cloud probe (i.e. the Cloud Droplet Probe, CDP), and two optical array probes (OAPs) for larger hydrometeors (D > 50 µm). The CDP (Lance et al., 2010) provided 5 Hz cloud droplet (1 to 50 µm) size spectra in bins 1 to 2 µm wide. The CDP RMS accuracy of mean droplet diameter of 0.7 µm was determined

after the campaign using the University of Wyoming droplet generator (Faber et al., 2018).

The OAPs, on the other hand, imaged larger hydrometeors (D > 50 µm) as the particles pass through an illuminated sample volume and shadow individual members of a linear photodiode array. The 2D Stereo Probe (2DS; Lawson et al., 2006) imaged particles at a 10-µm resolution across a 1.28 mm diode array, accurately resolving the hydrometeor spectra for particles 50 µm < D < 1 millimeter. The 2D Precipitation Probe (2DP) measured hydrometeors larger than a millimeter with an image resolution

of 200 µm. The data from the OAPs were processed using the University of Illinois OAP Processing Software (Jackson et al., 2014, Finlon et al., 2016), to perform standard image rejection and dimension corrections. Size distributions were produced from image-derived size and particle timing information and a calculated sample volume—estimated from particle diameter, true airspeed, laser wavelength, and particle acceptance criteria following Heymsfield and Parrish (1978). Shattering by large ice crystals was avoided using antishattering tips on the 2DS and by filtering of particles with a short, static interarrival time

threshold in the software processing.

From these size distributions, several integrated water content metrics were calculated to estimate the instantaneous mass distribution within certain drop size categories of interest. The total—i.e. across the entire measured liquid hydrometeor size spectrum—LWC was integrated from the combined CDP and 2DS size spectra under the assumption of no SLW drops larger than the 1.3 mm upper size limit of the 2DS, based on visual inspection of probe images. The Cloud Water Content (CWC)

and Drizzle Water Content (DWC) metrics contain the mass from the 2-50 µm and 50 µm-1.3 mm parts of the cloud hydrometeor spectrum, respectively and hence sum to $LWC_{tot}$. The calculated $LWC_{tot}$ was compared to the bulk estimate from the Rosemont icing probe (also sensitive to SLW drops of D > 50 µm) over two mostly liquid legs from the research flight to validate these estimation methods. The only remarkable disagreement between the metrics came for LWC values of the Rosemont above 0.4 g/m$^3$, where the integrated $LWC_{tot}$ became larger compared to the Rosemont icing probe measurement.

This overestimation may be related to mis-sized particles over 50 µm in diameter (i.e. imaged, drizzle-sized donuts and partial donuts), corroborated by a similar departure of the Rosemont from the integrated CWC for these same high LWC values, suggesting that the disagreements were caused by drizzle-sized particles (missed in the CWC and overestimated in the $LWC_{tot}$).



While other potential sources of measurement error exist (particularly for the Rosemont probe), both the estimates integrated from the CDP and 2DS for these high LWC points err in the direction suggestive of mis-sizing of drizzle drops, making it the
likely error source.

The following results and analysis produced from the WCR profiles, in-situ bulk probes, and cloud microphysics datasets from the first UWKA flight in SNOWIE, highlights the role of sub-kilometer vertical velocity fluctuations on the spatiotemporal distribution of SCDDs and the inferred cloud microphysical response.

## 3 Results

The results presented are from the period of 0245 to 0405 UTC (legs 1, 2, and 5) during the first flight of the field campaign on January 7-8, 2017. Two distinct layer clouds developed in the wake of a precipitating frontal cloud system. Of these two clouds, the elevated cellular cloud layer contained both low background number concentrations of ice and cloud droplets and embedded kilometer or longer regions of SCDDs that formed in a larger pattern of orographic lift.

### 3.1 Synoptic and Thermodynamic Context

The UWKA research flight followed the passage of a deep snow band associated with a weak jetstreak in the 500 mb wind field (not shown). The deep, saturated atmosphere present in the upstream sounding during the heavily-precipitating period (roughly 4 hours prior to leg 1 start; Fig. 2a), experienced mid-tropospheric drying, and veering and strengthening of the winds above 8 km MSL. This led to lowered cloud tops and a pronounced dry slot from 7 to 9 km in the pre-flight sounding (~45 min prior to leg 1 start, Fig. 2b). This dry layer contained thin layers of expected dynamic instabilities—defined by bulk
Richardson number from 0 to 0.5 (Fig. 2b; blue shading). The layer below, between 4 and 7 km, saw several vertical humidity variations accompanied by evaporational cooling of the radiosonde upon exiting cloud layer tops, resembling conditional instabilities (orange shading). These layers were not expected to correspond to real convective motions in cloud.

By the start of the first flight leg at 2:45 UTC, a shallow orographic cloud layer persisted over the Payette basin on the western end of the flight track, with cloud tops around 4 km MSL (Fig. 3a)—matching the top of the lower saturated layer in the pre-
flight sounding (Fig. 2b). This orographic cloud layer was capped on the eastern end by a layer of broken, cellular cloud structures roughly 1-3 km wide—hereafter the elevated cellular layer—resembling, at times, either coherent K-H billows or incoherent generating cells. This elevated cellular layer was consistently strongest (in terms of layer depth and highest radar reflectivities) over the highest terrain at the east end of the leg.

The final upstream sounding (Fig. 2c; ~1 hour after leg 1 start) indicated a deeper saturated layer through 6.5 km and further
strengthening and veering of the wind above, with more vertically homogeneous, near-zonal winds between 3 and 6 km. This shear profile resulted in several layers of indicated dynamic instabilities within the 500 m above and below the top of the saturated layer and matched the 6 to 6.5 km cloud tops in flight legs 4 and 5 (nearest in time; Fig. 3d/e).





Variations in humidity and wind, superimposed on the background zonal winds and low-level orographic clouds, appeared responsible for an elevated cloud layer that was at times dynamically unstable and variable in vertical location and depth (Fig.

2b). Additionally, a surface inversion and attendant low-level static stability was present in all the upstream soundings around the time of the flight (Fig. 2a-c). As a result, calculated bulk Froude numbers were consistent with blocked flow below 2 km MSL (not shown), matching the overall low-level static stability pattern of the entire field campaign (Tessendorf et al., 2018). The stability from this surface inversion may have helped to decouple the surface airmass from the free troposphere above the Payette Mountains barrier.

**3.2 General Cloud Structure and Vertical Motions**

There were several differences between the orographic cloud layer (4.5 km MSL and below) and the cellular layer above. The orographic cloud layer persisted over the nearest 1-2 km above the terrain, with cloud tops that rose slightly (no more than 500 m) from west to east with the average height of the topography beneath (e.g. Fig. 3a). The cellular layer, however, was transient—discrete layers of cells advected into the target area at varying altitudes. Some of these layers appeared coupled to

the lower orographic cloud layer (as in legs 1, 2, 4, and 5), while others appeared totally separate (as in legs 3, 9, and 10). This behavior is consistent with the large vertical variations in wind shear and humidity between the three soundings in this layer (Fig. 2), including several dynamically unstable layers. Consistent with this, several of the elevated layers appeared to contain overturning (or breaking) cells in the reflectivity profiles, within the elevated cellular layer of leg 4 from 10-15 km downwind of PJ (Fig. 3d).

Across the entire research flight, the upper cloud layer maintained reflectivities less than -5 dBZ outside of individual fall streaks, which remained discrete as they advected across the flight track. This behavior suggested mostly liquid cloud species in the elevated layer, confirmed by the 99th percentile of precipitation-sized ice number (integrated from the 2DP probe) for each of the first four legs remaining below 0.1 $L^{-1}$ (leg 5 was only marginally higher, with a 99th percentile value of 0.3 $L^{-1}$). Some of the higher reflectivity fall streaks (especially towards the end of the flight) may have corresponded to seeding lines

(French et al., 2018; Tessendorf et al., 2018; Hatt et al., 2019) after the seeding period started at the end of leg 2, but do not warrant any further consideration beyond noting the location and effect of ice on observed reflectivities and size distributions. The reflectivities in the lower orographic cloud layer, by comparison, were greater than in the cellular layer above, with whole sections of the nearest 1 km AGL of cloud above 5 dBZ, suggesting ice below the orographic cloud top (or interface). The inferred relative abundance of ice in this shallow orographic layer may be due to more abundant aerosol (and INP) presumed

to reside below the strong surface inversion (Fig. 2b) or from secondary ice multiplication in the warm (-5 < T < -15 °C) temperatures in the lower layer, but no direct measurements were available in cloud below flight level.

Mean reflectivity-weighted, near-vertical Doppler velocities (hereafter, hydrometeor vertical velocities or Doppler velocities) were available from the WCR to quantify cloud vertical motions (i.e. the convolution of vertical air motions and reflectivity-weighted population terminal velocity). Additional corrections were applied to remove the contributions by the horizontal

wind for profiles where the radar beams deviated from vertical. Unfortunately, the complex dynamics down to sub-kilometer




scales convoluted with hydrometeor size and phase inhomogeneity confounded the observed Doppler velocities, making assumptions about a constant hydrometeor fall speed specious. In fact, the spread of fall speeds associated with observed hydrometeor size and phase variations—from the negligible fall speeds of populations of cloud droplets to the 1 m/s or more fall speeds of drizzling populations—were greater than the spread of air motions observed in the dynamic structures of focus

(< 1.5 m/s amplitude where sampled at flight level).

Despite this complexity, there were several obvious and consistent trends in the observed Doppler velocities: nearly all legs showed a distinct terrain-induced vertical velocity couplet centred roughly 24 km downwind of Packer John and directly above a pronounced N-S ridge, oriented perpendicular to the mean wind and flight direction (Fig. 4). This couplet consisted of up to 2 m/s upward Doppler velocities over the upwind slope immediately followed by as much as 4 m/s downward Doppler

velocities on the downwind side, and frequently extended up to cloud top (as in leg 5). Despite the wave-like signatures present in the reflectivity profiles, Doppler velocity couplets (away from flight level) and phase relationships (at flight level) between perturbation kinematic and thermodynamic quantities (not shown) were inconsistent with K-H waves. For this reason, care was taken separately in (1) quantifying the effects of hydrometeor fall speed spatial variation and (2) adopting the label of semi-coherent vertical velocity fluctuations (SCVVFs) to distinguish layers of these regularly-spaced, oriented vertical velocity

perturbations from the more isotropic turbulent motions found elsewhere. Probable meteorological sources for SCVVFs in this environment include K-H waves, shear-driven mechanical overturning (Houze and Medina, 2005), and shallow convective overturning with some regular triggering mechanism; however, the actual sources did not seem to uniquely affect the microphysics and therefore remain undistinguished.

### 3.3 Comparisons Between Drizzling Legs (1, 2, and 5)

The three legs of interest, legs 1, 2, and 5 (Table 2), were flown from 3.9-4.5 km MSL, each encountering kilometer-or-longer stretches of SCDD measured at flight level within the elevated cellular cloud layer, with significantly larger drops on the first two legs despite similar cloud water contents across all three. These regions were all located at or downwind of Packer John mountain (the start of prominent terrain features along this transect), where reflectivities and cloud layer thicknesses were consistently near the leg maxima (leg 4 being the lone exception). Above the windward slope of the Sawtooth Range, from

10-25 km downwind of PJ, was a broad region of ascent observed on most legs (0-1 m/s hydrometeor upward velocities) which contributed to the relatively high reflectivities and cloud layer thicknesses compared to cloud further upwind (Fig. 4). From 10-60 km downwind of PJ (the regions of interest for SCDDs), flight level vertical velocities for the three legs varied from -0.5 to 2 m/s, with perturbation magnitudes on legs 1 and 2 of up to 0.6 m/s and entirely lower than 0.2 m/s for leg 5 (Table 2). These legs sampled altitudes from 3900-4500 m, corresponding to temperatures as low as -16 °C (leg 2 5) to -11 °C (for the

lower altitude leg 5).

The sampled Cloud Water Content (CWC) values measured by the CDP were very similar for these drizzling sections of cloud, with maximum values approaching 0.6 g/m³ in both legs 1 and 5—the more widespread SCDD extent and occasionally broken cloud conditions of leg 2 only saw CWC as high as 0.4 g/m³, possibly reduced due to scavenging and removal of cloud water



by drizzle in the time between legs 1 and 2 (Table 2). Cloud droplet number concentration for these legs remained below 35

cm$^{-3}$, the observed maxima for this research flight, and decreased to values lower than 5 cm$^{-3}$ within portions of cloud with

significant SCDD sedimentation from above (as in legs 1 and 2), where both cloud and drizzle drops appeared to be the largest.

These SCDD plumes contained as much as 1 g/m$^3$ liquid water distributed over drizzle sizes (i.e. DWC, D > 50 μm). Total

spectral MVD's for these SCDD plumes approached 80 μm (Table 2). Unlike the first two legs, the SCDDs sampled in leg 5

were much smaller, with MVD only approaching 45 μm even within the plumes.

The primary microphysical differences for these three legs were the smaller SCDDs in leg 5 relative to legs 1 and 2. A further

cloud kinematic structural difference is the focus of the following section.

### 3.4 Semi-Coherent Vertical Velocity Fluctuations

The primary structural difference between the elevated cellular cloud layer for these three legs, which appeared responsible

for differences in cloud droplet size and SCDD vertical level of development, were the presence and vertical location of layers

of semi-coherent vertical velocity fluctuations (SCVVF's). A train of these velocity fluctuations were sampled at flight level

during the first leg and illustrate the cloud microphysical response (Fig. 5). Here, from 24 to 35 km downwind of PJ, the

SCVVF's appeared as a series of ± 0.5 m/s vertical velocity perturbations from the mean with roughly 1-2 km wavelength

(Fig. 5b). The vertical velocity fluctuations drove both a thermodynamic (Fig. 5e) and microphysical response (Fig. 5c/d),

which saw positive perturbation vertical velocities paired with lower temperatures, higher cloud droplet number, and lower

CWC relative to the trend. Appreciable drizzle mass was only present in the perturbation downdrafts (Fig. 5c, pink curve).

When averaged size distributions were examined for individual perturbation up and downdrafts (Fig. 6), it was apparent that

secondary droplet activation was primarily responsible for the increased droplet number concentration. The averaged size

distributions corresponding to positive perturbation updrafts show that much of the increased droplet number concentration

can be explained by the large number of 6-8 μm droplets, which are an order of magnitude more abundant than in the

interspersed downdrafts and nearly as abundant as droplets in the primary mode from 25-35 μm. Given that these legs were

flown at a constant altitude, the secondary droplet activation in perturbation updrafts, paired with lower CWC than the trend,

may indicate kinetically limited parcel behavior and is examined in the discussion. The perturbation downdrafts contained

increased drizzle mass (i.e. DWC), larger droplets, and lower total number concentration relative to perturbation updrafts. The

decreased number and increased DWC are likely explained by scavenging by the larger drops, which were as large as 150 μm

(Fig. 6) and indicate active collision-coalescence processes at flight level. Furthermore, collision-coalescence likely began

very near above flight level, as the reflectivity values were between -25 and -15 dBZ within the nearest 400 m above flight

level, indicative of populations cloud droplets with very few if any drizzle drops (Fig. 5a).

Spatiotemporal profiles of Doppler velocity (Fig. 7) highlight the difficulty in identifying layers of SCVVF's away from the

aircraft using the WCR. Near flight level from 25-30 km, where the gust probes indicated a regular perturbation velocity

pattern with 1-2 km spacing (dashed line, Fig. 7b), there is no similar hydrometeor vertical velocity pattern in the nearest few

gates to the UWKA (Fig. 7a). For comparison, within the nearest 200 m to cloud top, between 30-35 km downwind of PJ, the



top of a clear train of these vertical velocity fluctuations can be seen (Fig. 7a). These Doppler velocity fluctuations match the crests of the wavelike reflectivity structures near cloud top in the corresponding reflectivity profile (Fig. 5a, top circled), but do not extend as far downward into cloud as the reflectivity structures. This perturbation velocity pattern is clearest in the
highest 200 m of cloud in part due to the smaller sizes and terminal velocities of populations of scatterers there, compared to the radar volumes containing drizzle drops below, where the Doppler velocities become gradually more negative as the drizzle drops begin to dominate the reflectivity and where reflectivity-weighted terminal fall velocities become greater than the air motions. A similar increase in reflectivity-weighted terminal velocities—this time estimated from comparing flight level gust probe and near-aircraft Doppler velocities—occurred at flight level 29 km downwind of PJ, where a sharp increase in estimated
fall speed is noted from SCDD falling from aloft. This matches the increase in drizzle mass and spectral MVD beginning at nearly the same time (blue arrow, Fig. 5c/e).

The link between SCVVF and hydrometeor growth was also apparent in the Contoured Frequency by Altitude Diagrams (CFADs) of WCR radar reflectivity. For the region corresponding to the sampled SCVVF train at flight level (25-30 km downwind of PJ; Fig. 8a), the median reflectivity rapidly increased from a roughly constant -25 dBZ above 5 km MSL (500
m above flight level) to higher than -15 dBZ just below flight level—consistent with a transition from cloud droplet (D < 30 µm) to drizzle drop sizes for the low (N < 35 cm$^{-3}$) number concentrations in these clouds. This increase was characterized by a roughly -20 dBZ/km slope in the reflectivity CFAD which appeared consistently with the layers of SCVVF's elsewhere in cloud this day, e.g. where a layer of these SCVVF's appeared at cloud top (~6 km MSL) in the next 5 km of cloud downwind (Fig. 8b). The reflectivity enhancement tied to both layers of SCVVF's was discrete, in comparison to the more gradual growth
that occurred furthest downwind on this leg, starting at cloud top and extending through the entire cloud layer (Fig. 8c).

The result of these layers of SCVVF's on the broader microphysical character of sampled cloud for leg 1 was a trend of increasing size with distance downwind. At the broad 0.5-1 m/s updraft from 20-25 km downwind of PJ (Fig. 5b), the cloud hydrometeor size spectrum resembled a population of strictly cloud droplets with diameters almost entirely smaller than 40 µm (outside of the overestimation in the 2DS curve beyond the discontinuity; Fig. 9a, red). In the region of SCVVFs
immediately further downwind (25-35 km downwind of PJ), the primary modal diameter shifts to larger sizes while the steep exponential tail toward large sizes simultaneously flattens out into drizzle shoulder (Fig. 9a, green and blue). Further downwind (Fig. 9a, orange and purple) of the SCVVF train, a mature drizzle shoulder (100 µm < D < 300 µm) becomes apparent from SCDD's falling from the layer near cloud top. These SCVVF layers appear to be responsible for the trend of increasing drop size with distance—as layers of SCVVF's formed over prominent terrain elements, hydrometeor growth was enhanced and
drop sizes increased below and downwind of these layers.

Leg 2 saw the SCVVF layers from leg 1 break down into incoherent turbulence between legs, with the elevated cellular layer containing a prominent drizzle precipitation plume from 45-53 km downwind of PJ, capped by a turbulent and variable cloud top height (circled, Fig. 10a). Still present were juxtaposed perturbation updrafts and downdrafts, especially near cloud top (Fig. 10b), but these were not well-organized or layered as in leg 1 and did not have a unifying spatial scale. The circled drizzle
plume in the reflectivity field (Fig. 10a) agreed with the 0.4 g/m$^3$ or higher DWC where the 0 dBZ and higher reflectivities





crossed flight level (Fig. 10d). While several short wavelength perturbations appeared in the flight level vertical velocity profile (Fig. 10c), they did not have a consistent effect on the either the thermodynamic (Fig. 10e) or bulk microphysical data (Fig. 10d), unlike leg 1.

Leg 5, by comparison, contained a longer, shallower layer of SCVVF's from 12-33 km downwind of PJ between 4.5-4.8 km
MSL and 500-1000 m below cloud top (Fig. 11a, circled). The horizontal scale of these fluctuations was smaller than in leg 1, with the width of a complete up/down perturbation couplet narrower than 1 km for this SCVVF train (Fig. 11b). Perhaps because of both the relatively thin layer of these SCVVF's and nearness to flight level (only 750 m above flight level), drops were much smaller and the spectrum MVD remained below 45 μm (Table 2). Averaged size distributions at flight level below these SCVVF's indicated mostly small drizzle drops with diameter just greater than 50 μm, some larger ice hydrometeors
toward millimetric sizes (Fig. 9c), and relatively even mass distribution between CWC and DWC (Fig. 11d), unlike legs 1 and 2. The presence of ice was corroborated by 2DS probe images (not shown) indicating that any vertical reflectivity enhancements from layers of SCVVF's for this leg are complicated by the increased linear growth rates (and hence reflectivity response) of ice in a mixed phase environment.

Reflectivity and Doppler velocity CFADs for three 5 km-wide drizzling columns from legs 1, 2, and 5 were generated for
comparison (Fig. 12). The incoherent turbulence at cloud top for leg 2, seen in the large spread of Doppler velocities in the highest 1 km of cloud (Fig. 12e), produced a similar vertical reflectivity enhancement as in the Eastern end of leg 1 (Fig. 8c), where reflectivity gradually increased with distance downward over the elevated cellular layer. This pattern also appears in drizzling marine stratocumulus clouds where drizzle production typically occurs at cloud top and drizzle drops grow throughout the entire cloud layer (e.g. Comstock et al., 2005). For both drizzling columns, the broadening processes associated
with incoherent turbulence and entrainment at cloud top were sufficient for drizzle production and subsequent collectional growth through the whole cloud layer. By comparison, the thin embedded layer of SCVVF's present in leg 5 led to a shallow growth layer with larger reflectivity-altitude gradients (i.e. more horizontal slope in the thinner shaded growth region; Fig. 12g) than in either legs 1 or 2. The larger ice particles present in the tail of the corresponding size distribution for the column from leg 5 (Fig. 9c; confirmed by 2DS images—not shown) explain the similar median radar reflectivity values (-5 to 0 dBZ)
crossing flight level between legs 2 and 5 (Fig. 12 d/g) despite the comparatively smaller, more numerous drizzle drops compared to legs 1 and 2. All three drizzling columns contained reverse S correlation patterns between reflectivity and Doppler velocity in the vertical, associated with hydrometeor growth and fallout over the layer (Fig. 12 c/f/i).

## 4 Discussion

Much of the previous work describing SCDD development in orographic, mixed phase cloud systems focused on the necessary
conditions for development—namely the low cloud droplet and ice number concentrations and sufficient condensate supply rates to support condensational growth to the droplet sizes required for active collision-coalescence (Rauber, 1992; Ikeda et al., 2007). Several other studies suggested conditions which may be responsible for accelerated drizzle development or for





relaxing these necessary conditions, introducing broadening mechanisms important for SCDD production in cloud (Pobanz et al., 1994; Korolev and Isaac, 2000). Of these, the relationship between fine wind shear levels, spatial supersaturation

fluctuations, and SCDD development has yet to be connected mechanistically by in-situ measurements, despite being identified both as associated with SCDD development (Pobanz et al., 1994) and, separately, as important for the spectral broadening seen in certain layer clouds (Cooper, 1989; Korolev, 1995; Korolev and Mazin, 1993). The observations here seem an important continuation of the work by Pobanz et al. (1994), which called for further airborne research investigating the link between layers of strong wind shear and SCDD development. While their explanation called for observations of K-H billows to

understand the production mechanisms, the microphysical behavior in layers of SCVVF's here seems to provide similar insight towards understanding these mechanisms.

**4.1 Microphysical Response to SCVVF Layers**

The insight provided from sampling one of these SCVVF trains with the in-situ cloud hydrometeor probes (Fig. 5) allows for some characterization of the microphysical processes in clouds of this type. Based on the flight level microphysical and

kinematic data, a conceptual model is presented to consistently describe the response to SCVVF layers (Fig. 13). The kinematic structure and LWC response for leg 1 saw positive (negative) perturbation updrafts (downdrafts) paired with negative (positive) LWC perturbations from the trend and positive (negative) number concentration perturbations associated with droplet activation (evaporation). For these regular vertical velocity fluctuations (and with sufficiently low $N_{CDP}$), the supersaturation response to vertical velocity fluctuations as described by Korolev (1995), is responsible for (re)activating interstitial CCN as

small (6-8 µm) droplets in the sub-adiabatic perturbation updrafts and separately broadening the bottleneck droplet mode from the repeated supersaturation fluctuations. Sub adiabatic implies LWC values below what is expected from the adiabatic LWC formulation,

$$LWC = \Gamma_{LWC} \cdot (z - z_{CB}), \qquad\qquad\qquad (1)$$

where $\Gamma_{LWC}$ represents the adiabatic lapse rate of liquid water content and is determined by cloud base temperature and pressure (Albrecht et al., 1990). The mean CWC for the SCVVF train seen at flight level was 0.25 g/m$^3$ with regularly spaced

oscillations ± 0.05-0.08 g/m$^3$ from that mean (Fig. 5c).

In a well-mixed (i.e. nearly constant equivalent potential temperature; Fig. 2), non-precipitating orographic layer cloud, the expectation is that for a constant altitude, the adiabatically-constrained LWC be nearly constant with only small perturbations the result of variation in the cloud base thermodynamic conditions, i.e. $P_{cb}$ and $T_{cb}$. Back of the envelope calculations estimate the specific adiabatic LWC lapse rate of this elevated cellular layer cloud to be around 0.001 g/m$^4$, taking the thermodynamic

conditions from the sounding at the interface between orographic and elevated cellular layers as a pseudo cloud base for this upper layer. Given the mean cloud water contents of 0.25 g/m$^3$ at flight level, this indicates roughly 250 m of ascent for the cloud parcels sampled at this altitude. Variations of ±5 °C at cloud base would then correspond to ±0.05 g/m$^3$ perturbations in LWC, and variations of ±50 mb would correspond to ±0.01 g/m$^3$ perturbations, respectively. While the orographic environment does predispose clouds to experience more variation in cloud base conditions than similar layer clouds associated with fronts





or boundary layers, cloud base thermodynamic variations of this magnitude are not expected at this spatial scale (0.5-2 km) and do not likely explain the regular CWC perturbation response. Instead, the perturbations of up to 40% of the mean CWC at a constant altitude were likely the result of dynamic or precipitation processes and not the cloud thermodynamics.

The primary effect on LWC—or, more aptly, CWC if only condensational effects are considered and where drizzle is not falling through parcels from above—due to cloud kinematics is the kinetic effect described by Korolev (1995). The negative

CWC perturbations in leg 1 were accompanied by local supersaturation sufficient for secondary droplet activation (i.e. $S>S^*>1$), inferred from the small droplet (6-8 µm) mode present in the averaged size distributions within these perturbation updrafts (Fig. 6a, red and blue curves). Such sub-adiabatic behavior seems linked to the kinetic limitation on condensational growth—cloud parcels had low enough ($N_{CDP} < 15$ cm$^{-3}$) droplet number concentrations that the "condensational inertia" of droplet populations in condensing out excess supplied water vapor governed the supersaturation response, associated CWC

response, and secondary droplet activation behavior. For the droplet populations below 30 cm$^{-3}$ and with mean count diameter of roughly 20-30 µm, the corresponding phase relaxation time is around 10 s (using estimation methodology by Fukuta and Walter, 1970; Polotivitch and Cooper, 1988; and Korolev, 1995). This phase relaxation time corresponds to expected perturbations from the adiabatic mean of as much as 0.02 g/m$^3$ at flight level which indicate that, while the kinetic effect cannot explain the full perturbation magnitude in the CWC field, it acts in the proper observed direction and explains the primary

adiabatic (i.e. closed parcel) effect in these clouds. It is important to note, that while CWC would be maximized at maximum parcel displacement for *instantaneous* condensation, the condensational inertia represents a spatiotemporal lag displacing these maxima (minima) into the perturbation downdrafts (updrafts), as illustrated in Fig. 13.

The remaining magnitude of CWC variation seems to be related to either the precipitation dynamics or the breakdown of the "well-mixed" assumption implicit in the vertically-stratified adiabatic cloud model. In the first case, removal of cloud water

by scavenging from drizzle in perturbation updrafts would lead to lower CWC's than expected from the kinetic-adiabatic model alone. Interspersed perturbation downdrafts see larger drizzle drops and more drizzle mass (consistent with the observed DWC pattern; Fig. 5c) and are likely the origin of drizzle fall streaks in the vertical. In the second case, if the perturbation velocity structure is sufficiently long-lived, for the long phase relaxation times here, the regular vertical velocity pattern may act to advect or deform the local vertical CWC stratification. In this case, at a constant altitude, observed perturbation updrafts

would contain lower CWC advected from below which has yet to mix out with surrounding parcels or adjust via condensation. In this case, the vertical CWC contour deformations required to explain the remaining 0.03-0.07 g/m$^3$ of CWC perturbation would be on the order of 30-70 m and require the kinematic pattern to persist for 1-3 minutes given the relatively weak perturbation vertical velocity magnitudes (±0.2-0.5 m/s)—which seems unrealistic.

## 4.2 Reflectivity-Inferred Hydrometeor Growth in SCVVF Layers

The comparisons between vertical reflectivity, Doppler velocity, and their cross correlation suggest two main microphysical behaviors within layers of semi-coherent vertical velocity fluctuations. The first is rapid, and often discrete, drop growth in the vertical tied to layers of vertical velocity fluctuations, not confined to cloud top. This vertical growth rate appears as large for





these SCVVF layers in leg 1 as for the drizzle production at cloud top in leg 2, with similar observed LWC's and liquid-ice mass distribution (unlike leg 5). The second behavior is a reverse S cross correlation pattern (cf. Vali et al., 1998) for these

layers of SCVVF's, irrespective of hydrometeor phase differences, which further corroborates the local hydrometeor growth and fallout tied to these vertical layers.

Layers of SCVVF's in legs 1 and 5 were responsible for vertical reflectivity enhancements similar in magnitude (~-20 dBZ/km) as produced by the drizzling cloud in leg 2 where layers of SCVVF's were not present. However, these SCVVF layers (especially in the relatively upwind cloud elements closer to PJ) were responsible for discrete layers of growth that were not

confined to cloud top (Fig.'s 9a/10g). This indicates that the vertical velocity fluctuations were likely responsible for the initiation of collision-coalescence and drizzle production and occurred faster than classical cloud top broadening mechanisms (i.e. turbulent entrainment, isobaric mixing, etc) which further downwind or later were sufficient for drizzle production at cloud top. This was most apparent in the transition between legs 1 and 2 from discrete growth at the level of these SCVVF's to growth over the entire layer, starting at cloud top, in leg 2. While only a qualitative observation, this warrants an examination

of SCVVF's in other cloud regimes where embedded shear or shallow layers of static instabilities may be responsible for the vertical initiation of the collision-coalescence process. Layers of SCVVF's may also be important in clouds where condensational growth and cloud top spectral broadening occurs too slowly for active warm rain production, although with the caveat that any condensational kinetic effects are bound to be smaller than reported here. This agrees with the observations of both Pobanz et al. (1994) and Korolev and Isaac (2005).

A distinct feature of the layers of semi-coherent vertical velocity fluctuations is the bimodal DSD with populations of large (D > 30 µm) and small (D < 10 µm) droplets of similar number, not present elsewhere in cloud. This small droplet mode does not contain much mass compared to the large drop mode, and collisions between the large and small droplets are likely inefficient (E ~ 1-3% for drops of these sizes in laminar flow; Rogers and Yau, 1996), but the effect of such numerous possible collision events (especially given the large fall speed separations) in a turbulent environment may be enough to break the colloidal

stability of the bottleneck large drop mode for a few lucky drops, such that subsequent self-collection within this mode becomes favored. Furthermore, repeated supersaturation variations driven by vertical velocity fluctuations have been shown via parcel models to produce a local broadening about the larger droplet mode (Korolev 1995). This broadening may provide enough fall speed separation for self-collection without the need for larger droplets to physically interact with the newly activated droplets and agrees qualitatively with increases in drop size and drizzle mass with distance downwind within the vertical velocity

fluctuation layers where parcels may be expected to have undergone more supersaturation fluctuations.

The second apparent phenomenon—a reverse S vertical cross-correlation pattern between reflectivity and Doppler velocity across these growth layers—further corroborates the drop growth in these layers of vertical velocity fluctuations. This pattern, where it appeared in drizzling coastal stratus (Vali et al., 1998), was suggested to be the result of upward transport of drizzle and dilution of downward moving parcels near cloud top (region of positive correlation) which transitioned to the dominance

of precipitation terminal velocity effects below (region of negative correlation). Here the same trend is present in leg 5 (Fig. 12g), where the very low background reflectivities (-25 dBZ) above the growth layer transition to rapid reflectivity increases





below 5 km MSL correlated with positive Doppler velocities (Fig. 12i). As the Doppler velocities become more negative below this layer (Fig. 12h), the pattern reverses to the falling drizzle (and ice) dominating the reflectivity signature—with strongly negative correlations between reflectivity and Doppler velocity. The strongly negative correlation between reflectivity and

Doppler velocity in this region is dominated by the terminal velocity-size relationship (e.g. $v_T \sim D^2$ for drizzle drops) where volumes with the largest particles have the most negative Doppler velocities and highest reflectivities. At the top of the growth layer, where the weaker positive correlation exists between reflectivity and Doppler velocity, it is important to consider both the contribution of hydrometeor terminal velocity and air motion to the observed Doppler velocities. For the populations just above the growth layer, terminal velocities for the large (D ~ 35 µ) bottleneck cloud droplets are much less than the magnitude

of the vertical velocity perturbations (±0.5-1.0 m/s) and therefore the Doppler velocity signal is dominated by relative air motions. This suggests that the regions of upward relative air motion are correlated with higher reflectivities near the top of these SCVVF layers, though without in-situ measurements nearer the top of these layers to indicate whether primarily a size or concentration effect. A more expansive conceptual model (cf. Fig. 13) would incorporate the vertical gradient of these growth and fallout effects across the SCVVF layer but was too conjectural without more penetrations through SCVVF trains

at different altitudes.

## 5 Conclusions

Low droplet number concentrations ($N_{CDP} < 30$ cm$^{-3}$) and precipitation-sized ice number concentrations ($N_{2DP} < 0.5$ L$^{-1}$) despite cold cloud top temperatures (T ~ -30 °C), provided favorable conditions for supercooled drizzle drop development in a postfrontal orographic layer cloud forming over the Sawtooth Mountains east of the Payette Basin. This cloud, while transient

and variable in vertical location and depth, consistently was strongest over the prominent terrain features downwind of Packer John mountain, and frequently contained layers of semi-coherent vertical velocity perturbations. Where present in the elevated cellular layer cloud, layers of SCVVFs were associated with local SCDD development in response to the kinematic perturbation pattern and rapid vertical reflectivity enhancements (-20 dBZ/km) from hydrometeor collectional growth. This drizzle production and growth occurred embedded within cloud and over relatively shallow layers before transitioning to

drizzle production at cloud top and growth over the entire elevated cellular layer cloud.

## Author Contributions

AM performed the analysis and prepared the manuscript. JRF contributed to interpretation of results and provided critical edits in preparing the manuscript.



**Acknowledgements**

Observations from and participation in the SNOWIE field campaign was funded through NSF grant AGS-1547101, with UWKA participation supported by NSF grant AGS-1441831. We would like to acknowledge the contributions of both Coltin Grasmick and Phil Bergmaier for both feedback on the ideas present herein and shared access of their IDL libraries used in several figures. Finally, the feedback and suggestions from the SNOWIE principle investigators and senior scientists (Sarah Tessendorf, Lulin Xue, Kyoko Ikeda, and Roy Rasmussen of NCAR; Katja Friedrich of CU Boulder; and Bob Rauber of the

University of Illinois) were invaluable in honing in on the important elements of this analysis.

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



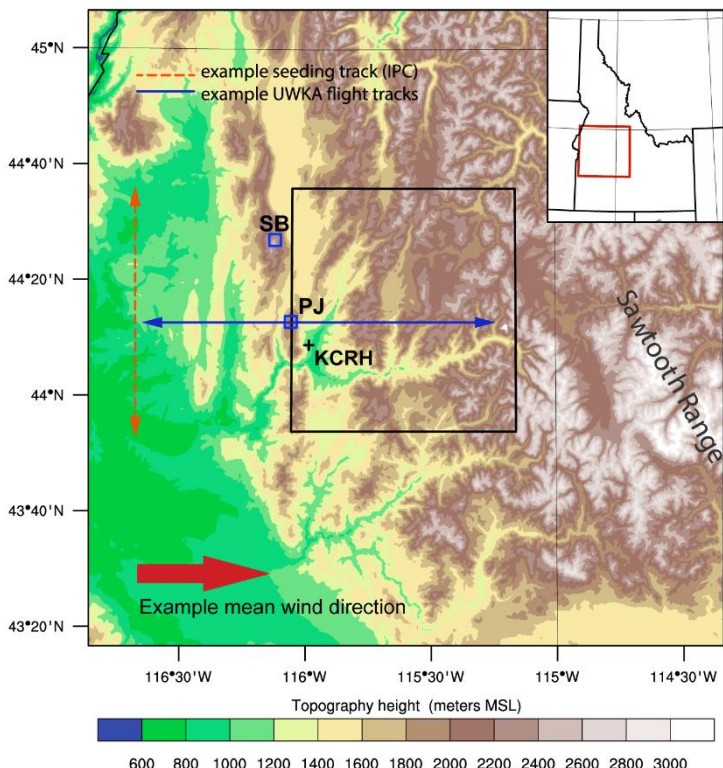

**Figure 1: SNOWIE plan view schematic for the example case of due westerly winds. Blue squares (▫) correspond to the Snowbank (SB) and Packer John (PJ) ground sites, the plus sign (+) indicates the Crouch (KCRH) sounding launch site. The rendered topography domain is the same as in orange (inset). The black bounding box indicates the target seeding domain.**






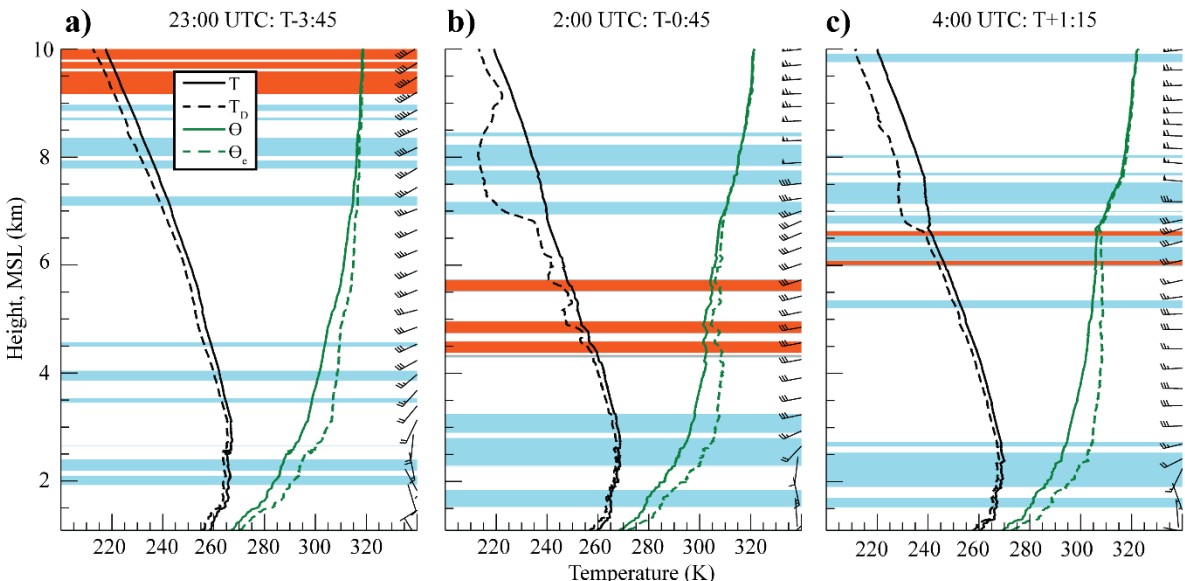


**Figure 2: Temporal development of vertical thermodynamic and dynamic profiles at the Crouch, ID sounding location (KCRH; Fig. 1). Shaded levels correspond to relaxed critical values of the bulk Richardson number, $Ri_{bulk} < 0.5$, after 10 pt (~50 m) vertical smoothing of the field. Orange shading indicates negative bulk Richardson values—corresponding to static instability—and blue corresponds to purely dynamic instability, $0 < Ri_{bulk} < 0.5$. Relative times (T +/-) reference the 2:45 UTC leg 1 start time.**






**Figure 3: Terrain-Referenced W-band Radar Reflectivity Spatiotemporal Profiles.** All distances are relative to Packer John Mountain, with positive (negative) values downwind (upwind). Leg start and end times are in UTC, with (a) through (j) corresponding to legs 1 through 10, respectively.




**Figure 4: Terrain-Referenced W-Band Mean Reflectivity-Weighted (Hydrometeor) Doppler Velocity Spatiotemporal Profiles.**
**Profiles have been corrected for aircraft attitude variations using sounding winds. All distances are relative to Packer John**
**Mountain, with positive (negative) values downwind (upwind). Leg start and end times are in UTC, with (a) through (j)**
**corresponding to legs 1 through 10, respectively.**





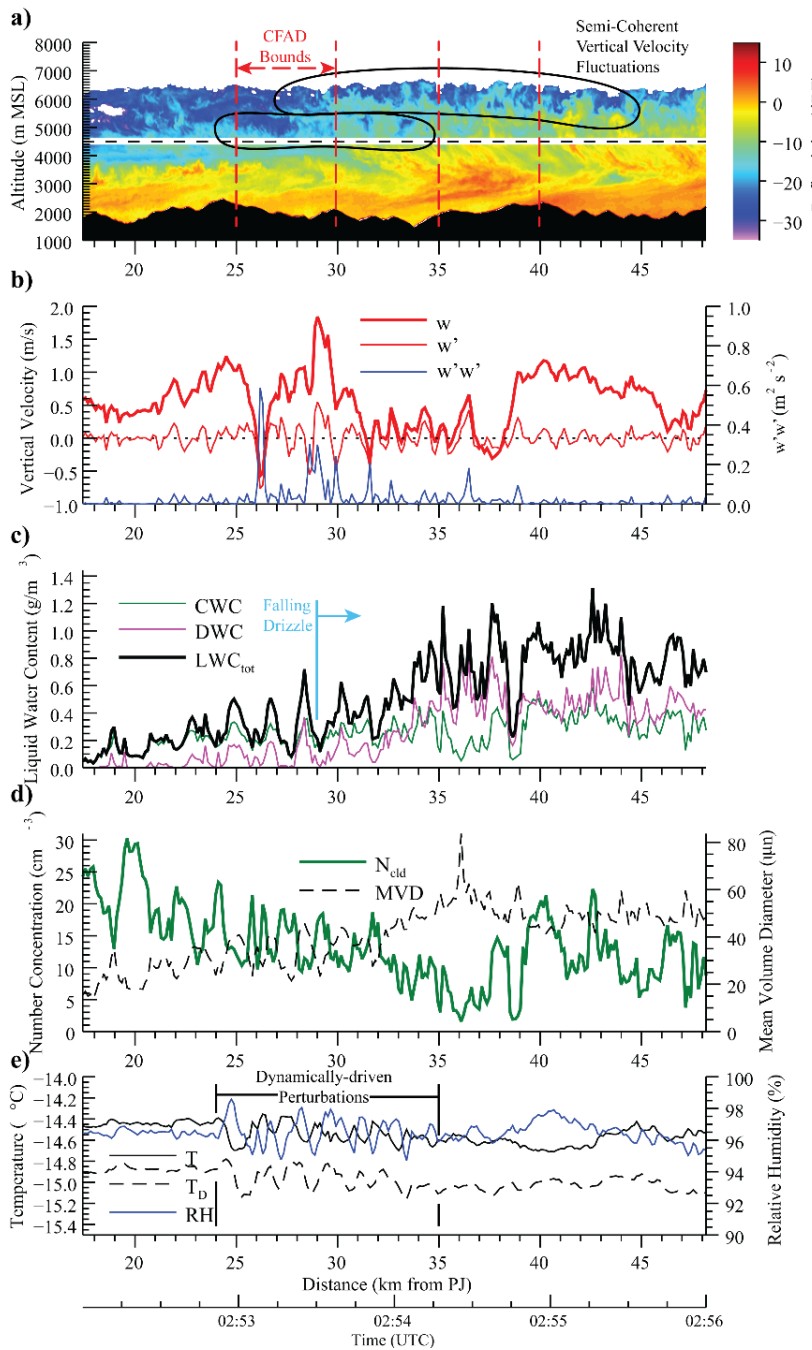

.

**Figure 5: Detailed radar and in-situ measurement profiles for the drizzling portion of leg 1. Spatiotemporal profiles of radar**
**reflectivity (a); actual and perturbation vertical velocity information (b); LWC measurements and integrated quantities (c); CDP**
**number concentration and combined (D < 1.2 mm) spectral MVD (d); and thermodynamic quantities (e). The CFAD bounds**
**correspond to the columns for Fig. 8a-c. Perturbation vertical velocities in (b) were calculated by subtracting a boxcar-smoothed**
**(over 10s or roughly 1 km) vertical velocity field from the actual gust probe vertical velocity and represent the sub-kilometer vertical**
**velocity perturbations.**





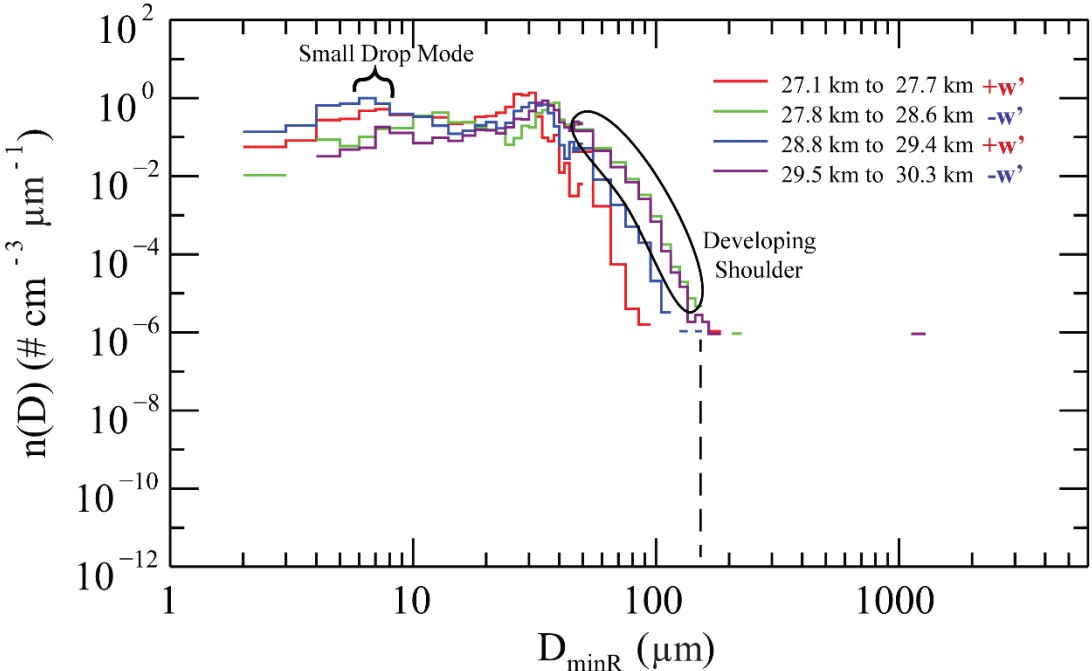

| Location | LWC$_{CDP}$ | LWC$_{comb}$ | N$_{CDP}$ | MVD$_{CDP}$ | MVD$_{comb}$ |
|---|---|---|---|---|---|
| *km* | *g m$^{-3}$* | *g m$^{-3}$* | *cm$^{-3}$* | *μm* | *μm* |
| **+w'** 27.1 - 27.7 | 0.155 | 0.182 | 16.988 | 25.950 | 27.238 |
| **-w'** 27.8 - 28.6 | 0.251 | 0.452 | 12.242 | 33.955 | 39.599 |
| **+w'** 28.8 - 29.4 | 0.177 | 0.217 | 15.528 | 27.897 | 29.676 |
| **-w'** 29.5 - 30.3 | 0.267 | 0.438 | 12.127 | 34.786 | 39.499 |

**Figure 6:** Bin-width normalized averaged size distributions for representative perturbation up-/down-drafts within the flight-level SCVVF train. Table (b) contains calculated distribution parameters for the curves in (a). Corresponding location downwind of Packer John located in top right legend, with location relative to perturbation vertical velocities by side.




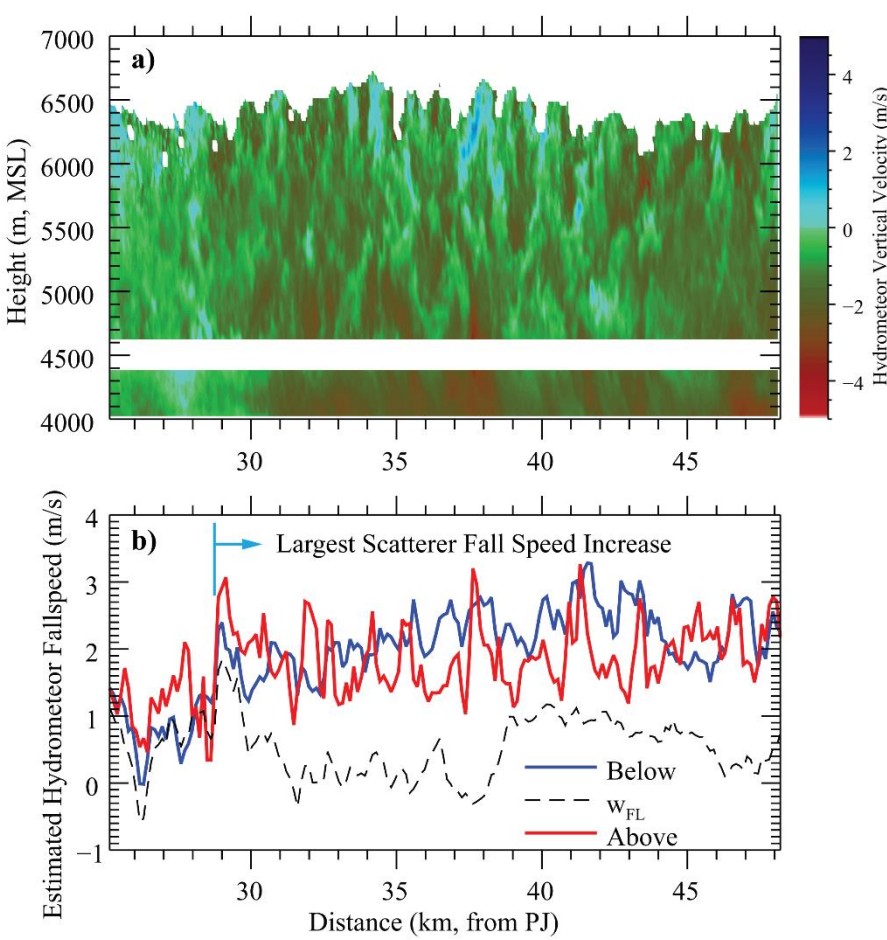

**Figure 7: Hydrometeor vertical velocity and estimated fall speed spatiotemporal profiles for Leg 1. Profiles of hydrometeor vertical velocity (a) and hydrometeor fall speed (b)—estimated by flight level gust probe vertical velocity (black dashed) minus averaged Doppler velocity of the 3 nearest useable radar gates above (red) and below (blue) flight level.**



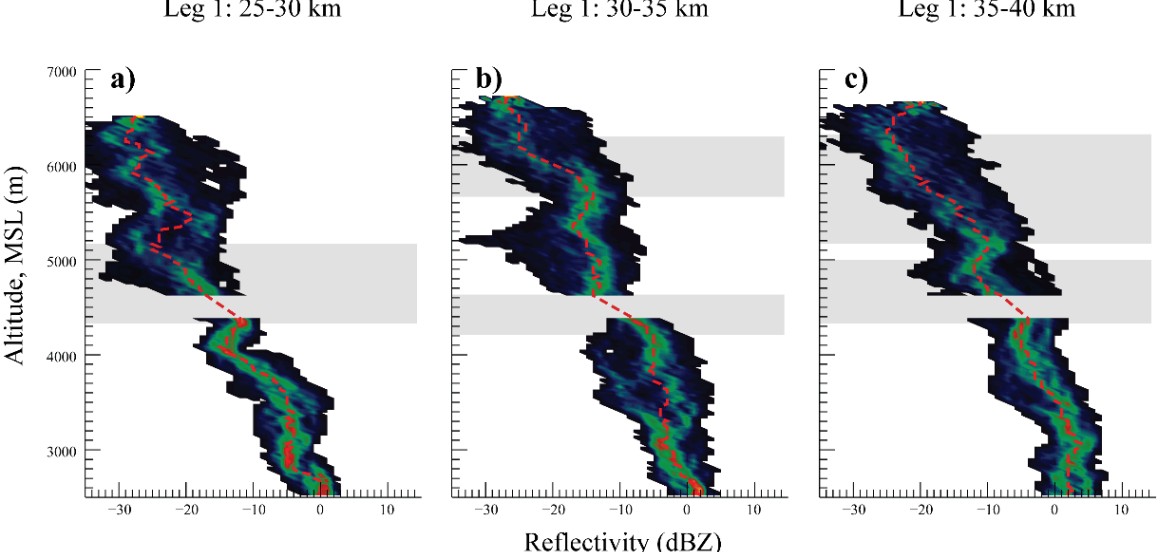

**Figure 8:** CFAD of radar reflectivity for three, 5 km-wide columns from leg 1, with relative location in km downwind of PJ at top. Dashed red line is median reflectivity for a vertical level and frequency is normalized for each vertical level (same colors at top as any other level). Shading indicates the primary inferred growth regions within the elevated cellular layer.





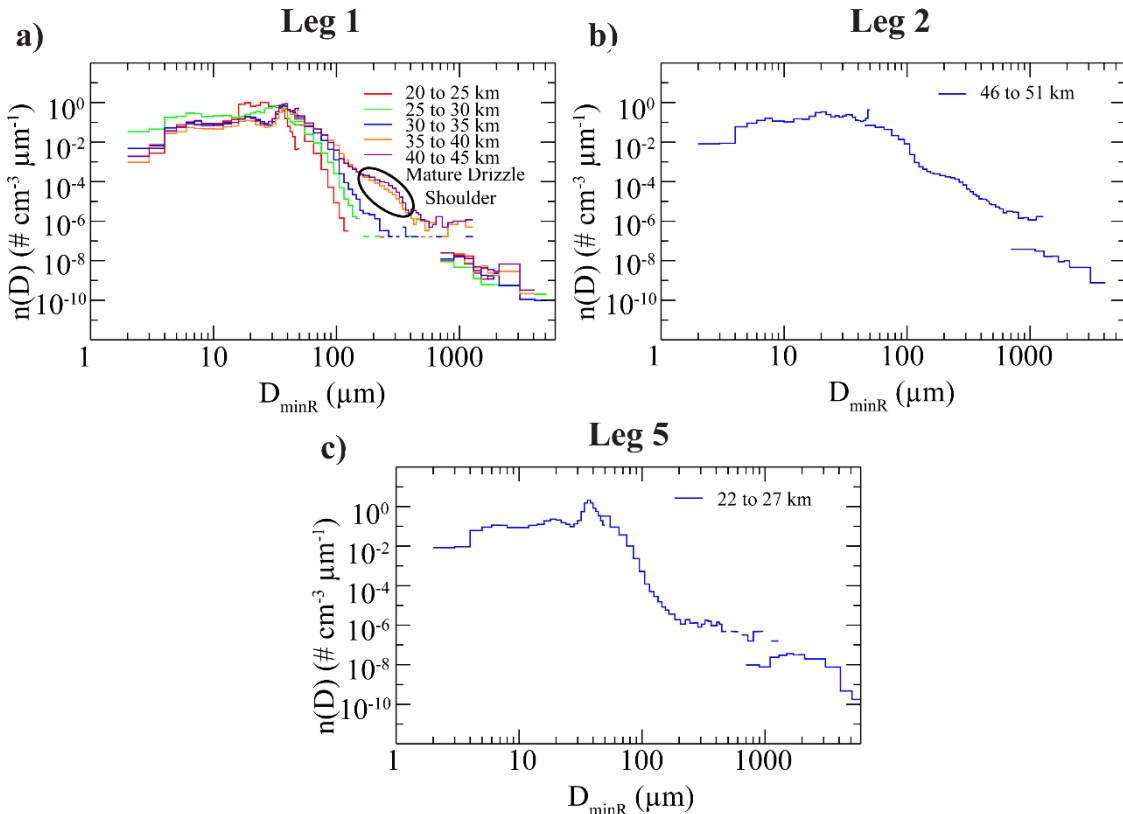

**Figure 9: Averaged size distributions for legs 1, 2, and 5 (a, b, and c respectively) from the CDP, 2DS, and 2DP cloud and precipitation probes. Each of the blue composite size spectra correspond to the averaged size distributions at flight level during the CFADs in Fig. 12.**





**Figure 10: Detailed radar and in-situ measurement profile for the drizzling portion of leg 2. Spatiotemporal profiles of radar reflectivity (a); Doppler velocity (b); flight level actual and perturbation vertical velocity information (c); LWC measurements and integrated quantities (d); and thermodynamic quantities (e). The CFAD bounds correspond to the column from Fig. 12d-f.**








**Figure 11: Detailed radar and in-situ measurement profile for drizzling portion of Leg 5. Spatiotemporal profiles of radar reflectivity (a); Doppler velocity (b); flight level actual and perturbation vertical velocity information (c); LWC measurements and integrated quantities (d); and thermodynamic quantities (e). The CFAD bounds correspond to the column from Fig. 12g-i.**








**Figure 12: CFADs of reflectivity, Doppler velocity, and their 0-lag cross-correlation for the legs 1, 2, and 5 (rows 1-3, respectively, with relative distances downwind of PJ indicated at top of row). Dashed red line (left column) is median reflectivity for a vertical level and frequency is normalized for each vertical level (same colors at top as any other level). Vertical profiles of 0-lag cross correlation between reflectivity and Doppler velocity in right-most panel with reverse-S correlation patterns highlighted in light blue. Shading indicates the primary inferred growth regions within the elevated cellular layer.**





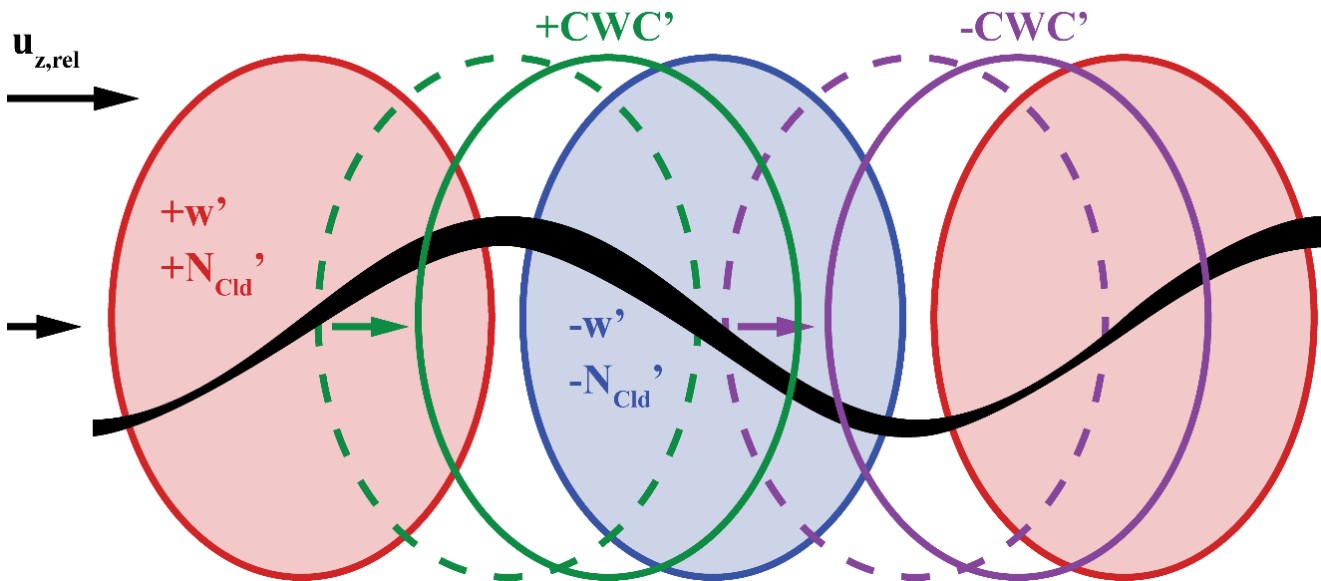

**Figure 13: Simplified schematic of spatial responses to the perturbation updraft (red) and downdraft (blue) pattern. The black trajectory indicates the approximate path and relative size of a droplet passing through the kinematic pattern. Cloud water content is expected to be maximized where parcel displacement is maximized between perturbation updraft and downdraft for instantaneous (dashed) condensation. The effect of a nontrivial condensational inertia shifts the perturbation CWC pattern downstream (arrows pointing to solid circles), translating the actual positive (negative) CWC perturbations into the perturbation downdrafts (updrafts).**




| Probe | CDP | 2DS | 2DP |
|---|---|---|---|
| Measured Sizes | 2 - 50 um | 5 - 1285 um | 0.4 - 16 mm |
| Sizing Technology | Forward Scattering | Optical Array | Optical Array |
| Temporal Resolution | 5 Hz | 1 Hz | 1 Hz |
| Approximate Spatial Resolution | 20 m | 100 m | 100 m |

**Table 1: Cloud Microphysics Probe Sizing and Technology**

| Leg | 1 | 2 | 5 |
|---|---|---|---|
| Altitude (m) | 4500 | 4800 | 3900-4200 |
| Temperature (°C) | -14.5 | -16 | -11 |
| Gust Probe Vertical Velocity (m/s) | -0.5 to 2 | -0.2 to 1.7 | -0.5 to 1.5 |
| Flight Level Perturbation Vertical Velocity Magnitude (m/s) | < 0.5 | < 0.7 | < 0.2 |
| Cloud Water Content (g/m$^3$) | < 0.6 | < 0.4 | < 0.6 |
| DWC in Plumes (g/m$^3$) | 0.2 to 0.8 | 0.1 to 1.0 | 0.1 to 0.4 |
| Cloud Droplet Number Concentration (cm$^{-3}$) | 2 to 30 | 3 to 30 | 8 to 35 |
| Mean Volume Diameter (μm) | < 80 | < 70 | < 45 |
| 99$^{th}$ Percentile Number Concentration of Precip-Sized Ice (L$^{-1}$) | 0.1 | 0.1 | 0.3 |

**Table 2: Flight Level Cloud Characterization Information Between Legs 1, 2, and 5**