# Peer review of "Supercooled Drizzle Development in Response to Semi-Coherent Vertical Velocity Fluctuations Within an Orographic Layer Cloud"

_Atmospheric Chemistry and Physics, 2019_

## Short Comment (SC1) · 16 Oct 2019

I came across this discussion manuscript via the weekly ACP alerts and found it to be very interesting. This manuscript has caught my attention because, in a manuscript of ours that was accepted for publication just last month (doi: 10.1029/2019JD030882), we describe a persistent highly supercooled drizzling event that was detected over McMurdo Station, Antarctica, during the recent AWARE field campaign. The drizzling cloud temperatures in our case study (-29 C < T < -25 C) were very similar to those reported in this study. I was happy to see that our estimations of the activated droplet

and ice number concentrations during that event based on LES simulations match the observational values in this study.

My only recommendation for the authors is to modify the Introduction sentence (l. 58-59): "... with few if any observations of SCDD formation found in the literature with cloud tops colder than -23 °C.", because there are, additionally, two previous reports mentioning highly supercooled drizzle observations at temperatures lower than -23 C, that is, Lawson et al. (2001; doi:10.1029/2000JD900789; T = $\sim$-25 C) and Korolev et al. (2002;Observation of drizzle at temperatures below-20 C. In 40th AIAA Aerospace Sciences Meeting & Exhibit; T = $\sim$-28 C).

Thank you, Israel Silber
* * *

---

## Author Comment (AC1) · 21 Oct 2019

Dr. Silber,

Thank you for your comment, and we are glad to see some agreement between your modeled number concentrations and our reported cloud droplet number concentrations. Additionally, the suggestion regarding other documented examples of supercooled drizzle observations below -23 °C in the literature is appreciated, and they will be included in the final publication.

[Figure]

Thanks, Adam Majewski

**ACPD**

---

## Referee Comment (RC1) · Anonymous Referee #1 · 13 Nov 2019

The paper uses insitu observations of dynamics and microphysics to link the formation of super cooled drizzle drops to specific dynamical conditions.

Main comment:

While the in depth interpretation of the flight leg data and back of the envelope discussion calculations were interesting, to me it seems that this qualitative investigation of this flight is the first part of the study. The hypothesis proposed is that scvvf are required to form scdd. I think it would be useful and necessary to generate some

quantification of the observations to test this hypothesis using the rest of the flight data available from the campaign.

Based on the qualitative hypothesis it would seem reasonable to try and define thresholds for the following conditions: 1. T<0 and T>Tmin 2. ice_conc < min_ice_conc 3. are scvvf present? Need some metric based on w' ? 4. are scdd present? Need some metric based on cloud probes.

and then combine these to quantify how well the scvvf-scdd hypothesis works. 1 and 2 have previously been suggested as controlling factors (as pointed out in the paper), while 3 is the new part explored here. So, if( 1 & 2 & 3) is true, is 4 also true? This can be assessed for different thresholds and metrics across the flight campaign. Such an approach could also be used to assess the frequency and usefulness of the S anticorrelation seen to be indicative of scdd. I think this level of quantification would be very useful for other researchers and have application in aviation safety.

Other comments:

l46 - S* and CCN not defined

l55 - and riming....

line 68 - what mechanism? Is it shear induced turbulent enhancement?

line 85 - what gradient? Number concentration with temperature or horizontal distance?

l151 - could report the frequency (Hz) of the data here for the size distributions, concentrations and condensed water estimates.

l212 - confirmed by the 2dp - was shape recognition used for the 2dp, or was the 99th percentile based on a size threshold?

l218 - 'suggesting ice ' - can liquid be ruled out? The doppler velocity would seem to be a potential evidence stream, but the text following this line seems to suggest it would be ambiguous.

l232 - it seems that the plots and analysis could be passed through a high-pass filter to remove the terrain induced larger scale fluctuations and just concentrate on the smaller scale variations.

l271 - okay fig5b has w', but it is not clear what the nature of the filtering was of w to derive w'.

l274 - to me the correlation between Ncld and w' looks poor - can you plot scatter plots and give correlation coefficients - or even do lagged correlations given the discussion at the end of the paper?

fig5d shows the mvd and Ncld to have an almost linear anticorrelation, suggesting that Ncld is proportional to LWC^(-0.5). I don't know if that is a coincidence or if it means something significant...

l332 - the hypothesis was posed earlier on that the SCVVFs were responsible for the SCDD, but now this observation seems to counter that. See my main comment above.

l351 - can you quantify this S correlation pattern to use it automatically?

l373 Ncdp is this the same as Ncld ?

l407 - is it possible to show a figure like 13 but from the actual data? I find it difficult to identify this behaviour in the current figures. Its difficult to see, but do the doppler velocity and reflectivity fields also show this lag effect?

---

## Referee Comment (RC2) · Anonymous Referee #2 · 1 Dec 2019

Review of "Supercooled drizzle development in response to semi-coherent vertical velocity fluctuations within a supercooled orographic layer cloud" by Majewski and French

This paper explores aircraft observations using a W-band radar and in-situ measurements of state parameters, wind motions, and cloud and drizzle drop size distributions within a supercooled layer cloud flowing over heterogeneous terrain. The authors find correlations between km-scale somewhat vertically coherent fluctuations in vertical motion and microphysical changes in the cloud. The measurements appear to support the idea that small scale fluctuations in such clouds may be sufficient to push an otherwise

non-drizzling cloud into a state whereby it produces precipitation.

I found the manuscript to be adequately well-written, although the authors should pay better attention to spelling (Rosemount is mis-spelled in several occasions), grammar, and precision in their scientific writing, The results, while interesting, are not particularly novel (see e.g. House and Medina 2005, who have already documented such correlations between small-scale vertical motions and radar-derived microphysical properties). It is a little unclear how the results presented would move our knowledge base forward. The authors need to work harder to make their results appealing to the broader cloud physics community.

The schematic diagram presented in Fig. 13 is interesting, but I believe that the condensational inertia theory of why the LWC and Nd estimates are not in quadrature is insufficient. As the authors argue, the condensational delay may be around 10 seconds (phase relaxation timescale), yet the time between wave crests is substantially longer than this (probably 100 s or more for wind speeds of 10-20 m/s and wavelengths of 1-2 km). More quantification of this would be helpful. Why isn't the removal of droplets by coalescence also playing a key role?

Why is there no map showing the synoptic conditions, horizontal flow pattern etc? Where are the Payette mountains?

I think the data here could be analyzed in a much more quantitative manner than is presented here. What is the vertical coherence of the small-scale vertical motions as seen by the WCR? This is why we have radars. Yet the radar here is underutilized.

Fig. 7 states that hydrometeor Doppler motions are show, but this implies that the vertical wind field is known. How can this be? This needs some correction to explain what is shown and what was done to remove the wind motions.

It is interesting that the clouds are ultra clean (very low cloud droplet concentrations). Yet this is barely mentioned later. Is there a real bottleneck for drizzle production given

this? The authors could quantify the coalescence by running an SCE solver on their size distributions to quantify the degree to which the clouds could produce drizzle without vertical motion enhancing LWC.

It is hard for me to understand why it important that the cloud is supercooled. Wouldn't the same physics affect warm layer clouds?

———————————————

---

## Author Comment (AC2) · 4 Feb 2020

**Reply to Referee #1**

Adam Majewski[1], Jeffrey R. French[1]

[1]Department of Atmospheric Science, University of Wyoming, Laramie, 82070, USA

*Correspondence to*: Adam Majewski (amajewsk@uwyo.edu)

We thank the reviewer for pushing us to better quantify the link between SCVVFs and SCDDs and pointing out a number of clerical errors. This, together with comments from the other reviewer led to a significant revision of the manuscript. We believe that the revised manuscript is easier to read, more consistent throughout, and provides conjecture and explanations that are better supported by the observations presented.

Below, comments provided by the reviewer are in black, our responses are in red.

**Reviewer Comments**

The paper uses in-situ observations of dynamics and microphysics to link the formation of supercooled drizzle drops to specific dynamical conditions.
Main comment: While the in depth interpretation of the flight leg data and back of the envelope discussion calculations were interesting, to me it seems that this qualitative investigation of this flight is the first part of the study. **The hypothesis proposed is that scvvf are required to form scdd**. I think it would be useful and necessary to generate some quantification of the observations to test this hypothesis using the rest of the flight data available from the campaign.

The bolded statement above mis-represents the intent of this manuscript. Through this work we aim to demonstrate that the presence of SCVVFs *enhance* drizzle formation and growth and therefore can influence where drizzle may form within clouds. This by itself is not surprising or novel. Throughout the introduction we present several previous studies linking the production of drizzle, in both supercooled clouds and in warm clouds, to atmospheric phenomena such as wind shear, turbulent mixing, cloud top instabilities, etc., which act to broaden the DSD. Further restrictions may be placed on *SCDD* development such as low CCN to grow larger droplets and INP concentrations to inhibit ice growth. The previous works focus on drizzle production predominantly at *cloud top*. Here we demonstrate that SCVVF layers can lead to drizzle initiation in the middle of a cloud, and need not occur at cloud top if the conditions are right. The observations presented from this case coupled with the concepts derived from earlier studies set up our hypotheses and eventual conclusions: that SCVVFs (1) can enhance collision-coalescence growth in a macroscopic sense (inferred from vertical reflectivity gradients), (2) can influence the vertical location of collision-coalescence onset in cloud (first occurring where SCVVFs are present), and (3) fundamentally affect the cloud microphysical response by condensation (subadiabatic w'-CWC' relationship). It is not our intent to suggest that SCVVFs are a necessary condition for SCDD production. In fact, we present observations from leg 2 showing the presence of drizzle despite the *absence* of SCVVFs. This is similarly true near the far eastern end of leg 1, where drizzle is present without SCVVFs. In these cases, drizzle initiation is presumably occurring near cloud top. Care has been taken in the revised manuscript to make this point more clear to the reader.

Based on the qualitative hypothesis it would seem reasonable to try and define thresholds for the following conditions: 1. T<0 and T>Tmin 2. ice_conc < min_ice_conc 3. are scvvf present? Need some metric based on w' ? 4. are scdd present? Need some metric based on cloud probes. and then combine these to quantify how well the scvvf-scdd hypothesis works. 1 and 2 have previously been suggested as controlling factors (as pointed out in the paper), while 3 is the new part explored here. So, if( 1 & 2 & 3) is true, is 4 also true? This can be assessed for different thresholds and metrics across the flight campaign. Such an approach could also be used to assess the frequency and usefulness of the S anticorrelation seen to be

indicative of scdd. I think this level of quantification would be very useful for other researchers and have application in aviation safety.

While we agree with the reviewer that such a study would be very interesting and useful, it is completely separate from the work presented here. The case-study approach used here aims to provide insight into the mechanisms important for influencing SCDD formation and growth. One expects that this in turn can be used to inform and validate future detailed modeling studies that aim to reproduce SCDD development in case clouds. A subsequent, campaign-wide examination of the role of SCVVFs in hydrometeor growth should help inform us regarding their overall role in SCDD formation in general, but provides little insight into the mechanism(s) responsible.

**Specific Comments**

l46 - S* and CCN not defined
The symbol S* is removed in the revised manuscript. CCN now defined in paragraph 3 in the Introduction: "(i.e. with lower numbers of cloud condensation nuclei; CCN)"
Care has been taken throughout the revised manuscript to ensure all symbols and abbreviations are defined.

l55 - and riming....
Riming has been included in the revised manuscript. We also note the following sentence captures this by stating: "....else ice will more rapidly scavenge the available vapor and cloud water." Paragraph 4 of Introduction in revised manuscript.

l68 - what mechanism? Is it shear induced turbulent enhancement?
Changed to explicitly indicate "turbulent broadening or mixing". Middle of paragraph 5 of the Introduction in revised manuscript

l85 - what gradient? Number concentration with temperature or horizontal distance?
This entire paragraph has been removed in the revised manuscript.

l151 - could report the frequency (Hz) of the data here for the size distributions, concentrations and condensed water estimates.
Change made, revised manuscript now reads: "From these 1 Hz size spectra…" in describing the size distributions and derived water content estimates. Paragraph 5, Section 2 of the revised manuscript.

L212: confirmed by the 2dp - was shape recognition used for the 2dp, or was the 99th percentile based on a size threshold?
The resolution (200 μm) of the 2DP is too coarse for reliable shape recognition. Rather, 2DP measurements were only used for particles with diameters greater than 1 mm (Paragraph 5, Section 2). Visual inspection of 2DS images failed to reveal any obvious liquid drops, *i.e. very circular particles,* with diameters larger than about 500 μm. Therefore, we presume that any particles larger than 1 mm, detected by the 2DP are likely ice. Regardless of whether these particles are liquid or ice, it does not change the conclusion that the concentration of ice particles was less than 0.1 $L^{-1}$ in legs 1 and 2 and 0.3 $L^{-1}$ in leg 5.

L218: "suggesting ice" - can liquid be ruled out? The doppler velocity would seem to be a potential evidence stream, but the text following this line seems to suggest it would be ambiguous.
We thank the reviewer for pointing this out. We often observed a significant decrease in Doppler velocity within about 1 km of the surface (Fig. 4e and 4f, for example, 20 to 40 km downwind of PJ). This decrease occurs at a similar location to a corresponding increase in the radar reflectivity, further bolstering the conjecture that liquid is being transformed to ice and that subsequent growth leads to enhanced reflectivity near the surface. This has been included in the revised manuscript. Paragraph 2 of Section 3.2.

L232: it seems that the plots and analysis could be passed through a high-pass filter to remove the terrain induced larger scale fluctuations and just concentrate on the smaller scale variations.

l271 - okay fig5b has w', but it is not clear what the nature of the filtering was of w to derive w'.

In order to calculate perturbation vertical velocity (i.e. *w'*), we took the measured vertical velocity and subtracted a simple high-pass filtered field that had been processed with a 10-s boxcar moving average. Several different size filters were tried, and the 10-s filter seemed to adequately capture the perturbations of interest. The details of this calculation are now included in the caption of Fig. 5 in the revised manuscript.

L274: to me the correlation between Ncld and w' looks poor - can you plot scatter plots and give correlation coefficients - or even do lagged correlations given the discussion at the end of the paper? fig5d shows the mvd and Ncld to have an almost linear anticorrelation, suggesting that Ncld is proportional to LWC^(-0.5). I don't know if that is a coincidence or if it means something significant...

Both reviewers commented on the correlations between droplet number concentration/liquid water content and perturbation vertical velocity. This also plays into the development of the conceptual model, presented as figure 13 in the manuscript. In order to address both reviewers, we provide additional analysis showing computed lagged correlations and follow-on discussion at the end of each review response (see below).

To the reviewer's other comment--there is clearly a strong anticorrelation between cloud droplet number concentration and mean-volume diameter, although what is not clear from the figure is whether it is linear. In the case of a linear (or near linear) relationship, we wonder if this demonstrates a balance between growth through condensation, collision-coalescence, and removal of drops through scavenging/collection and sedimentation. Exploring this in future work, particularly using detailed parcel model frameworks may be worthwhile.

l332 - the hypothesis was posed earlier on that the SCVVFs were responsible for the SCDD, but now this observation seems to counter that. See my main comment above.

We point out again, that the hypothesis is that SCVVFs may *enhance* drizzle production, but are not *required* (or responsible) for the initiation of SCDDs. However, in the example referenced here (leg 5), SCVVFs are indeed present and SCDDs are sampled at flight level. The principal difference in leg 5 compared to leg 1 is just that the SCVVF's are contained within a thin layer just above flight level. The SCVVFs in leg 5 are described in the second-to-last paragraph in section 3.4

L351: can you quantify this S correlation pattern to use it automatically?

That is an interesting question...given the observations from this single case study, it would not be possible to derive a quantification to be applied automatically. However, it may be worth exploring in a broader study that uses many cases (across the entire SNOWIE campaign, for instance) to look for a robust signal. Such an effort is beyond the scope of this study, but could be folded into a subsequent campaign-wide examination of SCVVFs as noted in an earlier response.

l373 Ncdp is this the same as Ncld?

All reference to Ncld and Ncdp has been removed from the text and replaced with the more explicit "cloud droplet number concentration" in the revised manuscript.

L407: is it possible to show a figure like 13 but from the actual data? I find it difficult to identify this behaviour in the current figures. It's difficult to see, but do the doppler velocity and reflectivity fields also show this lag effect?

Based on the results of the lagged correlations (below) this figure has been significantly revised. We acknowledge that the original figure was difficult to interpret and the connections between the ideas represented in the figure and the observations were not well represented. The revised figure is much simpler, and these connections are more apparent. We also note that the observed behavior supporting this model is best seen in the flight level data because of the noted complexity of the Doppler velocity data.

**Condensational Inertia and Conceptual Model**

**R1**: To me the correlation between Ncld and w' looks poor - can you plot scatter plots and give correlation coefficients - or even do lagged correlations given the discussion at the end of the paper?

**R2**: The schematic diagram presented in Fig. 13 is interesting, but I believe that the condensational inertia theory of why the LWC and Nd estimates are not in quadrature is insufficient. As the authors argue, the condensational delay may be around 10 seconds (phase relaxation timescale), yet the time between wave crests is substantially longer than this (probably 100 s or more for wind speeds of 10-20 m/s and wavelengths of 1-2 km).

Both referees indicated a desire for better quantification of the relationship between w' and CWC'/Ncld'. For referee 1 this had to do with some suggested Ncld/CWC relationship while for referee 2 it concerned the time/spatial scales and the lack of time series signals being in the expected quadrature relationship as per the proposed conceptual model (Fig. 13). To examine these relationships more quantitatively, the higher (5 Hz) resolution w', CWC', and Ncld' time series for were detrended and filtered of frequencies smaller than 0.1 Hz (wavelengths longer than 1 km). These time series were lagged by 0.2 s increments over a full 10 s period and correlated with a Pearson autocorrelation function to determine the correlation coefficients at each time lag. The results are presented below:

[Figure]

**Figure 1: Normalized perturbation time series for selected kinematic and microphysical measurements. Measurements have been de-trended and filtered of frequencies lower than 0.1 Hz (corresponding to wavelengths longer than ~1 km).**

[Figure]

**Figure 2: Lagged Pearson correlation coefficients for the time perturbation quantities in Fig. 1.**

The lagged correlations (with maxima at position 0) clearly indicate that these condensational kinetic responses are zero-lag, with w'-CWC' being anticorrelated and w'-Ncld' positively correlated. Zero-lag correlations in this context likely indicate that cloud parcels are moving with(in) the kinematic pattern as opposed to through it as the latter should result in some spatial lag corresponding to the motion of parcels relative to w' pattern. This analysis has led us to modify the conceptual model presented in the original conceptual model with a more simple one. The analysis, while discrediting the original model, otherwise strengthens the suggested microphysical response. Furthermore, for the phase relaxation time to have led to

a spatial lag exactly in quadrature now seems obviously unlikely, and we thank the referees for asking us to quantify these relationships. A new conceptual model incorporating this insight is proposed with perturbations caused from kinetic responses to a vertical velocity couplet more closely resembling the overturning cells suggested in HM05 to avoid the flow continuity issues that arise from this vertical parcel motion framework (e.g. when considering the location of maximum vertical displacement of parcels).

[Figure]

**Figure 3: Revised conceptual model: simplified schematic of spatial responses to the perturbation updraft (blue) and downdraft (red) pattern superimposed on broader orographic lift (broad blue arrow bottom). The colored trajectories indicate the approximate path of parcels passing through the kinematic pattern following the schema of Houze and Medina (2005). Lines of constant cloud water content (green) indicating the expected deformations due to condensational kinetic effects, with line weight corresponding to relative condensate mass. Cloud parcels circulate within the vertical velocity perturbation pattern and more and smaller drops are located in perturbation updrafts than downdrafts. CWC contours appear flat and unperturbed above and below the vertical velocity fluctuation pattern as they are determined by the adiabatic ascent in the broader uplift pattern.**

**References**

Houze, R. A. and S. Medina, : Turbulence as a Mechanism for Orographic Precipitation Enhancement, J.Atmos.Sci., 62, 3599-3599-3623, https://doi.org/10.1175/JAS3555.1, 2005.

---

## Author Comment (AC3) · 4 Feb 2020

**Reply to Referee #2**

Adam Majewski[1], Jeffrey R. French[1]

[1]Department of Atmospheric Science, University of Wyoming, Laramie, 82070, USA

*Correspondence to*: Adam Majewski (amajewsk@uwyo.edu)

We thank the reviewer for pushing to broaden the applicability and better articulate the novelty of the results and also for challenging us to better quantify the perturbation correlations, leading to a change to the conceptual model that seems to agree better with the flight data. This, together with comments from the other reviewer led to a significant revision of the manuscript. We believe that the revised manuscript is easier to read, more consistent throughout, and provides conjecture and explanations that are better supported by the observations presented.

Below, comments provided by the reviewer are in black, our responses are in red.

**Reviewer Comments**

This paper explores aircraft observations using a W-band radar and in-situ measurements of state parameters, wind motions, and cloud and drizzle drop size distributions within a supercooled layer cloud flowing over heterogeneous terrain. The authors find correlations between km-scale somewhat vertically coherent fluctuations in vertical motion and microphysical changes in the cloud. The measurements appear to support the idea that small scale fluctuations in such clouds may be sufficient to push an otherwise non-drizzling cloud into a state whereby it produces precipitation.
I found the manuscript to be adequately well-written, although the authors should pay better attention to spelling (Rosemount is mis-spelled in several occasions), grammar, and precision in their scientific writing.
Comments from both reviewers have led to a significant revision of the manuscript. In this revision, we have taken care to be more consistent with our wording, grammatically correct and consistent, and more precise in our descriptions. Because of this, we believe that the revised manuscript is easier to read and contains fewer inconsistencies that can lead to reader mis-understanding.

The results, while interesting, are not particularly novel (see e.g. Houze and Medina 2005, who have already documented such correlations between small-scale vertical motions and radar-derived microphysical properties). It is a little unclear how the results presented would move our knowledge base forward.
Houze and Medina (2005; HM05 for brevity) examined the enhancement of precipitation by turbulent overturning cells in coastal frontal systems with contained a significant orographic forcing. At relatively large spatial scales  (broad udrafts and large swaths of available condensate) and small spatial scales (~kilometer scale vertical motions embedded in layers of shear-driven overturning cells) orography was shown to modify the flow field to generate or otherwise enhance condensate supply rates, increasing upstream precipitation via increased collectional growth where condensate was locally concentrated. Although the cases analyzed were principally of precipitating mixed phase clouds with active ice nucleation processes, the authors suggested that for clouds with the 0 °C isotherm nearer the surface or with embedded bright bands, these turbulent overturning cells would be expected to similarly enhance growth rates for falling liquid hydrometeors. It is precisely in this context that several of our findings here are novel: (1) despite the w-LWC relationship reported in HM05 for mixed phase clouds (+w' and +LWC' on average), we found the opposite correlation which we believe to be a function of low droplet number concentration; (2) collectional growth still appeared to be enhanced through these layers despite the inverse w'-LWC' relationship; and (3) vertical location of initial collision-coalescence activity appeared to be tied to these layers even well below cloud top. For these reasons, this not only serves as a strong addendum to HM05 with respect to liquid or mostly

liquid clouds, but also raises questions as to whether turbulent motions, locally enhanced SLW pockets, or something else (e.g. condensational kinetic effects for liquid, lengthened trajectories for ice, etc) are responsible for the faster hydrometeor growth noted in these layers *for all conditions*.

The authors need to work harder to make their results appealing to the broader cloud physics community.
Care has been taken in the revised manuscript to place these results in the context of all highly supercooled liquid clouds (for instance, paragraph 7, Section 1; and paragraph 1, Section 5), liquid clouds with marine aerosol character (paragraph 5, Section 1), and to simplify the conceptual model as much as possible (paragraph 3, Section 4.1).

The schematic diagram presented in Fig. 13 is interesting, but I believe that the condensational inertia theory of why the LWC and Nd estimates are not in quadrature is insufficient. As the authors argue, the condensational delay may be around 10 seconds (phase relaxation timescale), yet the time between wave crests is substantially longer than this (probably 100 s or more for wind speeds of 10-20 m/s and wavelengths of 1-2 km). More quantification of this would be helpful.
Comments from both reviewers has led to additional analysis investigating correlations of perturbation quantities. This has resulted in a significant revision to the schematic diagram (Fig. 13) in the revised manuscript. Details of this are provided at the end of our comments.

Why isn't the removal of droplets by coalescence also playing a key role?
This is explicitly addressed in paragraph 4, Section 4.1: The remaining magnitude of CWC variation is likely related to the precipitation dynamics. Removal of cloud water by scavenging from drizzle in perturbation updrafts would lead to lower CWC's and reduced cloud droplet number."

Why is there no map showing the synoptic conditions, horizontal flow pattern etc?
We included no map of the synoptic conditions because we felt they were adequately described in the exposition of the case context, and that the bulk thermodynamic conditions were more enlightening in describing how and where clouds formed. Here we try to strike a balance between completeness and length of manuscript. Synoptic maps can be found in the Master's Thesis from which this manuscript was developed (Majewski 2019; Fig. 3.1, p. 55).

Where are the Payette mountains?
The Payette mountains are a locally-used reference to the western-most foothills of the broader Sawtooth Range. In the revised manuscript all reference to the Payette mountains has been removed and reference is now made to the Sawtooth Range, consistent with the map shown in Fig. 1.

I think the data here could be analyzed in a much more quantitative manner than is presented here. What is the vertical coherence of the small-scale vertical motions as seen by the WCR? This is why we have radars. Yet the radar here is underutilized.
Some attempt to quantify the coherence of the Doppler velocities and the reflectivities have been done in a statistical sense for the CFAD columns with the vertical profile of bulk correlation coefficients. However, going beyond this to investigate the coherence of small-scale motions is not trivial, given the convolved nature of what is measured by the Doppler radar. Variations in hydrometeor terminal fall speeds, especially for drizzle, are much larger than variations in vertical air motion.

Fig. 7 states that hydrometeor Doppler motions are shown, but this implies that the vertical wind field is known. How can this be? This needs some correction to explain what is shown and what was done to remove the wind motions.
Figure 7a shows the measured *Doppler velocity*, as indicated in the caption (note this is the same as shown in Figs. 4, 10b, and 11b). No attempt has been made to de-convolve the vertical air motion from the hydrometeor terminal fall speeds in any of these images. This is explicitly stated in the revised manuscript, at the end of paragraph 3, Section 2 and again in paragraph 3, Section 3.2.

Note that in Figure 7b, we *estimate* the hydrometeor terminal fall speed for range gates located near the aircraft (both above and below flight level). This is done by subtracting the aircraft measured vertical air velocity from the radar measured Doppler velocity in these range gates. The description of this is found near the end of paragraph 3, Section 3.4.

It is interesting that the clouds are ultra clean (very low cloud droplet concentrations). Yet this is barely mentioned later. Is there a real bottleneck for drizzle production given this? The authors could quantify the coalescence by running an SCE solver on their size distributions to quantify the degree to which the clouds could produce drizzle without vertical motion enhancing LWC.
We agree. The very low droplet concentrations encountered on this day and throughout the field campaign were a very interesting (and surprising!) observation. We reference low droplet concentrations throughout the manuscript as we describe the microphysical characteristics of the observed cloud. These low droplet concentrations are critical regarding our understanding of the role of condensational inertia as pointed out in Section 4.1.

To address the reviewer's question as to whether there is truly a condensational "bottleneck" for cloud droplet numbers as low as those reported here, it seems the corresponding marine stratocumulus research regarding ultra clean layers (Wood et al., 2018; Kuan-Ting O et al., 2018) might be most relevant. Laminar veil clouds that detrain from marine cumulus can persist on the order of hours against very weak lift (1 cm s-1). While containing drop effective radii in excess of 20 µm, the persistence of these clouds (timescales on the order of hours) against such weak updrafts suggests weak sedimentation and little collision-coalescence activity else clouds would more quickly dissipate. Subsequent modeling results (Kuan-Ting O et al., 2018) indicate that little if any sedimentation and collision coalescence persists after parcels moved into the detrained quiescent layer. Finally, DSD solutions for marine aerosol populations in vigorous (cumulus) updrafts have already been demonstrated to asymptote to an upper effective radius below 20 µm with diminishing dispersion and spectral width magnitudes above cloud base for a polydisperse parcel model (Pinsky et al., 2014), indicating that without broadening and/or collision-coalescence mechanisms, there is a definite upper limit to the size of droplets produced through condensational growth alone. Regardless, we have removed the "bottleneck" vocabulary from the revised manuscript and just directly refer to a narrow, large drop, condensational mode.

It is hard for me to understand why it is important that the cloud is supercooled. Wouldn't the same physics affect warm layer clouds?
In short, yes, the same physics apply in warm cloud layers. But consider for a moment marine StCu. Such clouds are BL phenomena occurring over a flat surface. There appear few if any drivers for SCVVFs to occur within the middle of these clouds. In such cases, vertical velocity fluctuations that can act to enhance drizzle production will almost certainly be confined to cloud top and therefore it should not be surprising that drizzle initiation occurs at the top of these cloud layers.

However, the emphasis here on a supercooled cloud must consider that: (1) this cloud had extremely cold cloud tops (T~-30°C), which Demott et al. (2010) suggest should lead to high INP concentrations, so the near absence of ice is quite surprising; and (2) for supercooled mixed phase clouds to produce SCDD requires relatively few CCN and INP. This scenario all but requires that supercooled clouds be inefficient precipitators with most of the mass distributed in the SLW categories. This also means that nearly all supercooled drizzling clouds can be expected to respond in kind to SCVVFs/Overturning Cells, which have already been acknowledged to be nearly ubiquitous in an orographic environment.

**Condensational Inertia and Conceptual Model**

**R1**: To me the correlation between Ncld and w' looks poor - can you plot scatter plots and give correlation coefficients - or even do lagged correlations given the discussion at the end of the paper?

**R2**: The schematic diagram presented in Fig. 13 is interesting, but I believe that the condensational inertia theory of why the LWC and Nd estimates are not in quadrature is insufficient. As the authors argue, the condensational delay may be around 10 seconds (phase relaxation timescale), yet the time between wave crests is substantially longer than this (probably 100 s or more for wind speeds of 10-20 m/s and wavelengths of 1-2 km).

Both referees indicated a desire for better quantification of the relationship between w' and CWC'/Ncld'. For referee 1 this had to do with some suggested Ncld/CWC relationship while for referee 2 it concerned the time/spatial scales and the lack of time series signals being in the expected quadrature relationship as per the proposed conceptual model (Fig. 13). To examine these relationships more quantitatively, the higher (5 Hz) resolution w', CWC', and Ncld' time series for were detrended and filtered of frequencies smaller than 0.1 Hz (wavelengths longer than 1 km). These time series were lagged by 0.2 s increments over a full 10 s period and correlated with a Pearson autocorrelation function to determine the correlation coefficients at each time lag. The results are presented below:

[Figure]

**Figure 1: Normalized perturbation time series for selected kinematic and microphysical measurements. Measurements have been de-trended and filtered of frequencies lower than 0.1 Hz (corresponding to wavelengths longer than ~1 km).**

[Figure]

**Figure 2: Lagged Pearson correlation coefficients for the time perturbation quantities in Fig. 1.**

The lagged correlations (with maxima at position 0) clearly indicate that these condensational kinetic responses are zero-lag, with w'-CWC' being anticorrelated and w'-Ncld' positively correlated. Zero-lag correlations in this context likely indicate that cloud parcels are moving with(in) the kinematic pattern as opposed to through it as the latter should result in some spatial lag corresponding to the motion of parcels relative to w' pattern. This analysis has led us to modify the conceptual model presented in the original conceptual model with a more simple one. The analysis, while discrediting the original model, otherwise strengthens the suggested microphysical response. Furthermore, for the phase relaxation time to have led to a spatial lag exactly in quadrature now seems obviously unlikely, and we thank the referees for asking us to quantify these relationships. A new conceptual model incorporating this insight is proposed with perturbations caused from kinetic responses to a vertical velocity couplet more closely resembling the overturning cells suggested in HM05 to avoid the flow

continuity issues that arise from this vertical parcel motion framework (e.g. when considering the location of maximum vertical displacement of parcels).

[Figure]

**Figure 3: Revised conceptual model: simplified schematic of spatial responses to the perturbation updraft (blue) and downdraft (red) pattern superimposed on broader orographic lift (broad blue arrow bottom). The colored trajectories indicate the approximate path of parcels passing through the kinematic pattern following the schema of Houze and Medina (2005). Lines of constant cloud water content (green) indicating the expected deformations due to condensational kinetic effects, with line weight corresponding to relative condensate mass. Cloud parcels circulate within the vertical velocity perturbation pattern and more and smaller drops are located in perturbation updrafts than downdrafts. CWC contours appear flat and unperturbed above and below the vertical velocity fluctuation pattern as they are determined by the adiabatic ascent in the broader uplift pattern.**

**References**

DeMott, P. J., A. J. Prenni, X. Liu, S. M. Kreidenweis, M. D. Petters, C. H. Twohy, M. S. Richardson, T. Eidhammer, and D. C. Rogers, : Predicting global atmospheric ice nuclei distributions and their impacts on climate, Proc.Natl.Acad.Sci.USA, 107, 11217-11217-11222, https://doi.org/10.1073/pnas.0910818107, 2010.

Houze, R. A. and S. Medina: Turbulence as a Mechanism for Orographic Precipitation Enhancement, J.Atmos.Sci., 62, 3599-3599-3623, https://doi.org/10.1175/JAS3555.1, 2005.

Kuan-Ting, O., R. Wood, and C. S. Bretherton: Ultraclean Layers and Optically Thin Clouds in the Stratocumulus-to-Cumulus Transition. Part II: Depletion of Cloud Droplets and Cloud Condensation Nuclei through Collision–Coalescence, J.Atmos.Sci., 75, 1653-1653-1673, https://doi.org/10.1175/JAS-D-17-0218.1, 2018.

Majewski, A. J., 2019: Supercooled Drizzle Drop Development In a Postfrontal Orographic Layer Cloud in Response to Semi-Coherent Vertical Velocity Fluctuations, University of Wyoming. https://search.proquest.com/docview/2282179118

Pinsky, M., I. P. Mazin, A. Korolev, and A. Khain: Supersaturation and diffusional droplet growth in liquid clouds: Polydisperse spectra, J.Geophys.Res.Atmos., 119, 12,872-12,872-12,887, https://doi.org/10.1002/2014JD021885, 2014.

Wood, R., K. O, C. S. Bretherton, J. Mohrmann, B. A. Albrecht, P. Zuidema, V. Ghate, C. Schwartz, E. Eloranta, S. Glienke, R. A. Shaw, J. Fugal, and P. Minnis: Ultraclean Layers and Optically Thin Clouds in the Stratocumulus-to-Cumulus Transition. Part I: Observations, J.Atmos.Sci., 75, 1631-1631-1652, https://doi.org/10.1175/JAS-D-17-0213.1, 2018.